# ADAM19 cleaves the PTH receptor and associates with brachydactyly type E

Atakan Aydin[1,2] , Christoph Klenk[3] , Katarina Nemec[1,3,7], Ali Işbilir[1,3], Lisa M Martin[1], Henrik Zauber[1], Trendelina Rrustemi[1], Hakan R Toka[2] , Herbert Schuster[2], Maolian Gong[2], Sigmar Stricker[4] , Andreas Bock[1,6], Sylvia Bähring[2] , Matthias Selbach[1] , Martin J Lohse[1,5], Friedrich C Luft[1,2]

Brachydactyly type E (BDE), shortened metacarpals, metatarsals, cone-shaped epiphyses, and short stature commonly occurs as a sole phenotype. Parathyroid hormone-like protein (PTHrP) has been shown to be responsible in all forms to date, either directly or indirectly. We used linkage and then whole genome sequencing in a small pedigree, to elucidate BDE and identified a truncated disintegrin-and-metalloproteinase-19 (ADAM19) allele in all affected family members, but not in nonaffected persons. Since we had shown earlier that the extracellular domain of the parathyroid hormone receptor (PTHR1) is subject to an unidentified metalloproteinase cleavage, we tested the hypothesis that ADAM19 is a sheddase for PTHR1. WT ADAM19 cleaved PTHR1, while mutated ADAM-19 did not. We mapped the cleavage site that we verified with mass spectrometry between amino acids 64–65. ADAM-19 cleavage increased $G_q$ and decreased $G_s$ activation. Moreover, perturbed PTHR1 cleavage by ADAM19 increased ß-arrestin2 recruitment, while cAMP accumulation was not altered. We suggest that ADAM19 serves as a regulatory element for PTHR1 and could be responsible for BDE. This sheddase may affect other PTHrP or PTH-related functions.

## Introduction

Metacarpal and metatarsal shortening, coupled with cone-shaped epiphyses and short stature, is termed brachydactyly type E (BDE) (Riccardi & Holmes, 1974). When BDE is inherited, an autosomal-dominant pattern is invariably observed. We identified a cis-regulatory site that downregulates *PTHLH*, the gene encoding parathyroid hormone-related peptide (PTHrP), as responsible for BDE (Maass et al, 2010). Subsequently, *PTHLH* gene deletions and point mutations were also shown to be responsible (Klopocki et al, 2010). In other BDE families with a balanced translocation, we identified a misplaced lncRNA disrupting the cis-regulatory landscape causing downregulation of *PTHLH* (Maass et al, 2012). In still a further balanced-translocation BDE family, we encountered a 2q37 deletion that included HDAC4 (Maass et al, 2018). As a result, differential expression of several genes was affected, including *PTHLH*, *SOX9*, and the lncRNA we had identified earlier. The above families all exhibited autosomal-dominant BDE with reduced stature as the sole phenotypes. However, we also described kindreds with BDE accompanied by severe arterial hypertension and stroke along with other vascular phenotypes (Schuster et al, 1996). In these families, we showed that mutated overactive phosphodiesterase 3A is responsible (Maass et al, 2015). *PTHLH* in these families is dysregulated, causing BDE. PTHrP is recognized as a major, locally expressed paracrine regulator of growth-cartilage chondrocyte proliferation, differentiation, synthetic function, and mineralization (Terkeltaub et al, 1998). Here, we investigated another family with solely the inherited BDE phenotype, as described earlier (Cartwright et al, 1980). The BDE could not be explained by any of the hitherto-fore described genetic mechanisms. Our linkage and sequencing data implicate mutated disintegrin and metalloproteinase-19 (ADAM19). The membrane-anchored ADAM enzymes are involved in numerous biological processes involving cell-cell and cell-matrix interactions (Mochizuki & Okada, 2007).

Earlier, our group had shown that the parathyroid hormone (PTH) receptor 1 (PTHR1) can be cleaved by an unidentified metalloprotease (Klenk et al, 2010a). PTHR1 also serves as the receptor for PTHrP (Suva & Friedman, 2020). PTHR1 is a family B, G protein-coupled receptor (GPCR) that regulates skeletal development, bone turnover, and mineral ion homeostasis (Cheloha et al, 2015). Our present work addresses the relationship between ADAM19 and PTHR1. We believe our findings are also relevant to the BDE

---

[1]Max Delbrück Center for Molecular Medicine in the Helmholtz Association (MDC), Berlin, Germany    [2]Experimental and Clinical Research Center, A Cooperation Between the Max Delbrück Center for Molecular Medicine in the Helmholtz Association and Charité Universitätsmedizin, Berlin, Germany    [3]Institute of Pharmacology and Toxicology, University of Würzburg, Würzburg, Germany    [4]Institute of Chemistry and Biochemistry, Freie Universität Berlin, Berlin, Germany    [5]ISAR Bioscience Institute, Munich, Germany    [6]Rudolf-Boehm-Institute of Pharmacology and Toxicology, Medical Faculty, University of Leipzig, Leipzig, Germany    [7]Department of Structural Biology and Center of Excellence for Data-Driven Discovery, St. Jude Children's Research Hospital, Memphis, TN, USA

Correspondence: friedrich.luft@charite.de

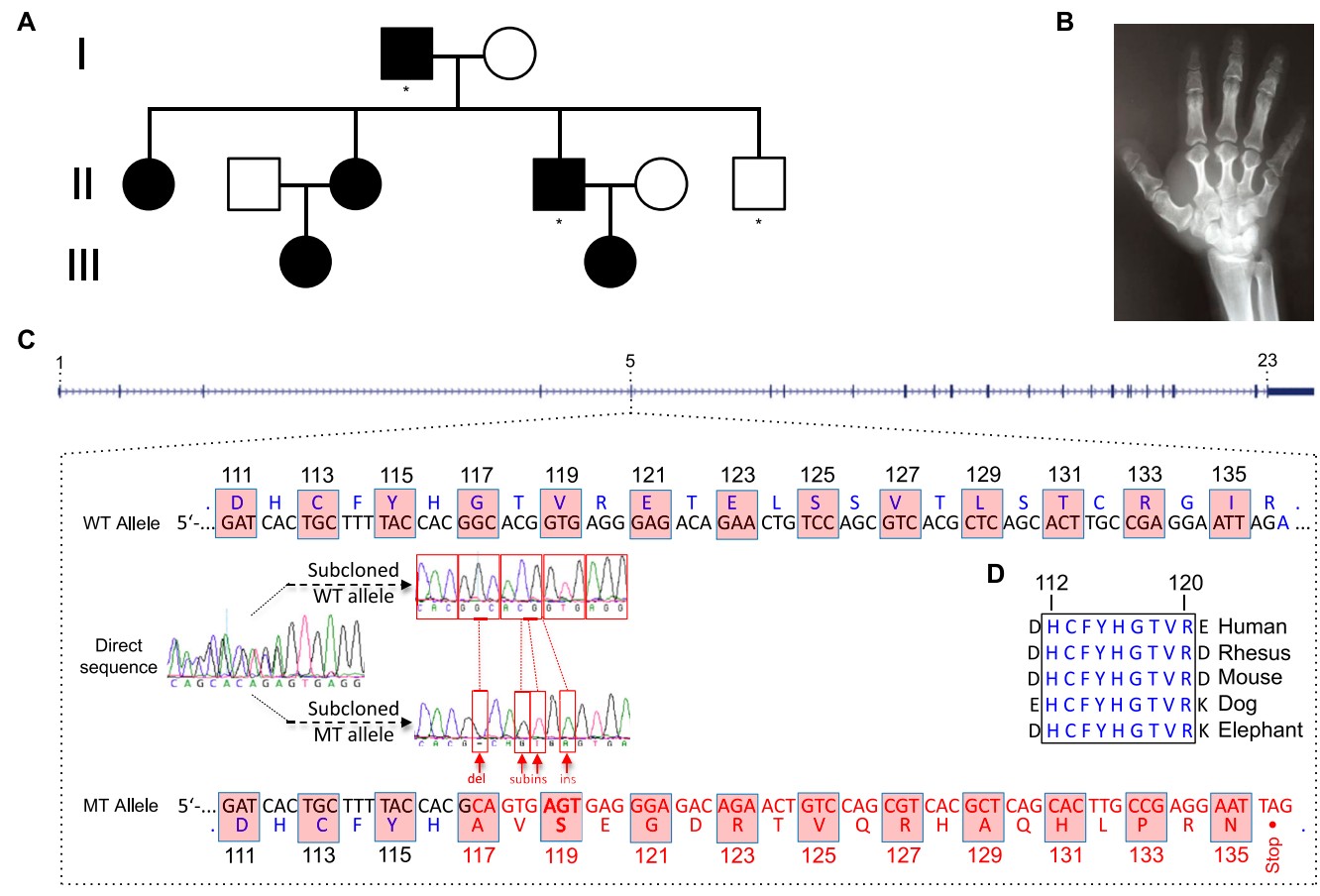

**Figure 1.  Family tree and mutations.**
The genomes of three family members were sequenced with the complete Genomics platform. All affected members have the ADAM19 mutation, while non-affected persons do not. **(A)** Sequenced persons designated with asterisk (A). **(B)** Hand roentgenograms of a patient illustrating brachydactyly type E with shortened, stubby metacarpals, cone-shaped epiphyses, and shortened proximal phalanges, which were present in all affected family members (B). *ADAM19* contains 23 exons. The mutation resides within exon 5. We show the amino acid sequence from amino acids 111–135. The mutation features a deletion resulting in an amino acid exchange at 117, followed by a substitution-insertion and a second insertion. As a result, amino acids 117–135 are faulty, compared to WT. **(C)** Finally, a TAG stop codon resides after amino acid 135 (C). **(D)** Multiple ADAM19 peptide-sequence alignments from five different species near the identified mutation(s) were highly conserved region (D).

phenotype and putative PTHrP signaling. Since PTHrP is intimately involved in how cancer metastasizes to bone, we suggest that the relevance of our findings extend far beyond a relatively uncommon genetic syndrome (Ponzetti & Rucci, 2020).

## Results

We revisited the BDE family described by Cartwright et al earlier (Cartwright et al, 1980). We examined 10 members, six of whom had the BDE phenotype (Fig 1A). A hand roentgenogram from an affected person shows shortened metacarpals and cone-shaped epiphyses (Fig 1B). In our clinical assessment, we identified no further phenotypes that were not shown earlier; we found no dental phenotypes, and none were previously mentioned. We first performed a linkage analysis with a 900 K single nucleotide polymorphism (SNP) array and detected genomic regions on seven different

chromosomes with LOD scores above 1.5. We performed fine mapping with microsatellite markers and identified promising loci (Fig S1A). Indel length and substitution length are given (Fig S1B and C). We then relied on whole-genome sequencing of two affected and one non-affected person from our kindred (Fig S1B). Of the four candidate loci only one, on chromosome 5, contained a gene, *ADAM19*, exhibiting the same mutation in both affected subjects (Fig 1C). This mutation was then identified in all BDE subjects and in none of the non-affected persons (Fig S2). The mutation features a deletion resulting in an amino acid exchange at 117, followed by a substitution-insertion and a second insertion. As a result, amino acids 117–135 are faulty, compared to the WT amino acid sequence. Finally, a TAG stop codon resides after amino acid 135. Multiple ADAM19 peptide-sequence alignments from five different species near the identified mutation indicate that the region is highly conserved (Fig 1D). We believe that in the PDF, the image is not great. The electropherograms are barely discernible and the text in the

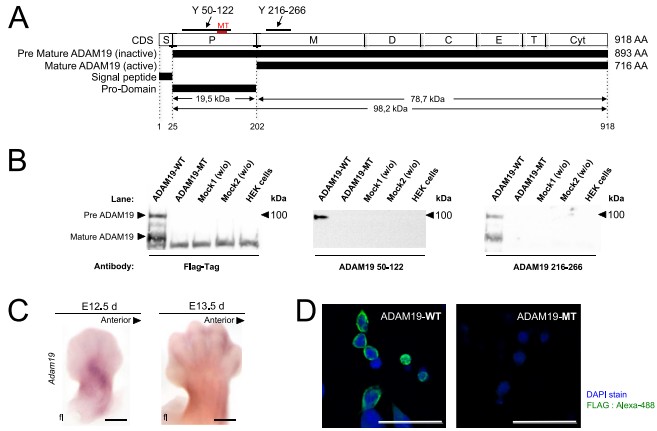

**Figure 2. Domain organization and ADAM19 structure.**
**(A)** The coding sequence contains 918 amino acids (AA) and different domains: signal peptide (S); pro-peptide domain (P); metalloproteinase domain (M); disintegrin domain (D); cysteine-rich domain (C); EGF-like domain; transmembrane domain (T); and the cytoplasmic (Cyt) domain (A). Immunoprecipitation results from HEK-293T cell extracts were performed. Full-length WT ADAM19 (WT; NM_033274.4) and ADAM19 with the mutations (MT) were transiently transfected in HEK-293T cells to express FLAG-tagged ADAM19 and analyzed with an antibody raised against the pro-domain (Y 50–122) with the epitope to amino acid 50–122, the metalloproteinase domain (Y 216–266) with the epitope to amino acid 216–266 of ADAM19 and a FLAG antibody directed at the tagged cytoplasmic-domain. Two mock transfection controls were used: mock1 (w/o) without ADAM19 cDNA but only with vector-DNA alone and mock 2 (w/o) without PEI. **(B)** Chondrogenic *Adam19* expression in mouse embryos was investigated at 12.5 and 13.5 d by RNA in situ hybridization. **(C)** *Adam19* was expressed in inter-digital space and the developing joints. Scale bars 1 mm (C). Flag-tagged ADAM19 was expressed in HEK-293T cells. **(D)** Immunofluorescence against the Flag-tag shows that WT ADAM19 is located on the membrane surface, while mutated ADAM19 was not identified (Scale bars 50 μm) (D).

image is not readable. We believe that the image we sent should be good enough to make this better.

The *ADAM19* coding sequence contains 918 amino acids (Fig 2A). The gene encodes for a signal peptide, a pro-peptide domain, a metalloproteinase domain, a disintegrin domain, a cysteine-rich domain, an EGF-like domain, a transmembrane domain and a cytoplasmic domain. The mutated protein was predicted to lack the transmembrane-to-cytoplasmic domain region. We next performed immunoprecipitation experiments (Fig 2B). Full-length WT ADAM19-FLAG-tag and mutated ADAM19-FLAG-tag were transiently transfected in HEK 293 cells. A FLAG-tag was fused in frame to the cytoplasmic domain of WT ADAM19 and specific antibodies directed against the pro-domain, the metalloproteinase domain, and the FLAG-tag were employed. The data showed detection of full-length WT ADAM19 and supported premature termination of mutated ADAM19. Mutation of ADAM19 leads to a frame shift resulting in a scrambled sequence after His116 and a new stop codon after amino acid 135. We performed Western blot experiments with antibodies against residues 50–122 of ADAM19 and with antibodies against the Flag epitope, which had been fused to the native C-terminus of ADAM19. The results suggested that neither the N-terminal fragment, nor full-length ADAM19 (e.g., as a result from a read-through event of the new stop codon), were formed and thus defective ADAM19 possibly has been degraded in the cell.

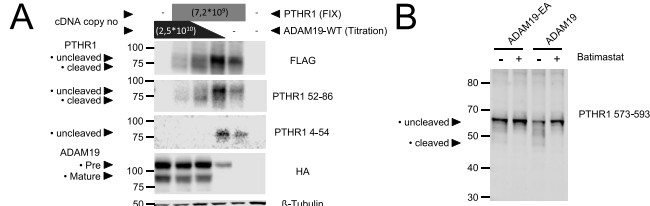

**Figure 3. ADAM19-shedding detection assay.**
For Western blot analysis, HEK-293T cells were transiently transfected for 48 h with ADAM19-HA-tag and PTHR1-FLAG-tag. PTHR1 was detected using antibody directed against the FLAG-tag, the amino acid epitope 52–86, and the amino acid epitope 4–54. ADAM19 was detected using an antibody directed against the HA-tag. **(A)** β-tubulin served as loading control (A). Verification of cleavage with native receptor and metalloproteases expressed in HEK293T cells for 36 h. Cells were lysed, and receptor proteins were de-glycosylated with PNGase F prior to analysis by SDS–PAGE and Western blot using a polyclonal antiserum directed against the amino acid epitope 573–593 of human PTHR1. **(B)** ADAM19-E384A is the inactive ADAM19 mutant (B).

However, we cannot fully exclude the possibility that the absence of ADAM19 at protein level may also be due to a pre-translational event.

We also studied 12.5-d-old and 13.5-d-old mouse embryos. These embryos expressed *Pthlh* as shown earlier (Maass et al, 2015) in the paw developing cartilage and also expressed *Adam19* (Figs 2C and S3). We performed immunofluorescence in HEK 239 cells transfected with WT and mutated ADAM19-FLAG-tag. We found that ADAM19 was strongly expressed on the plasma membrane together with PTHR1, while mutated ADAM19, was not detected (Figs 2D and S4). In addition, we assessed ADAM19 mRNA expression in various fetal tissues. We found that ADAM19 is particularly strongly expressed on chondrocytes and placenta. The findings are compared to cardiac expression as reference tissue (Fig S5).

We next studied whether ADAM19 is a PTHR1 cleavage enzyme. PTHR1 is a GPCR with seven transmembrane domains that we outfitted for Western blot analysis with a FLAG-tag at the C terminus and for fluorescence detection with an EYFP tag at the N terminus. We transfected HEK 293 cells with ADAM19-HA-tag and PTHR1-FLAG-tag. PTHR1 was detected using an antibody to the FLAG-tag, the amino acid epitope 52–86, and the amino acid epitope 4–54. ADAM19 was detected using an antibody to the HA-tag. The findings showed that WT ADAM19 cleaved PTHR1 (Figs S6 and 3A). To more precisely discriminate receptor species, proteins were deglycosylated by PNGase F prior to analysis. When co-expressed with catalytically inactive ADAM19 mutant (ADAM19-E384A), PTHR1 mainly appears as a band at ~60 kD, which is consistent with the molecular weight of the full-length receptor protein. In the presence of WT ADAM19, two additional bands at ~50 kD became apparent with increasing amounts of protease, while the 60 kD band was reduced. These findings suggest that a 10 kD fragment was lost at the receptor's N terminus in the presence of catalytically active ADAM19. This result would be in line with our previous findings that PTHR1 is cleaved by metallo-proteases within loop1 of the extracellular domain (ECD) (Klenk et al, 2010a, 2022). In contrast, when cells were cultivated in presence of 10 μM batimastat, a potent metalloprotease inhibitor, only the 60 kD species of PTHR1 were detected.

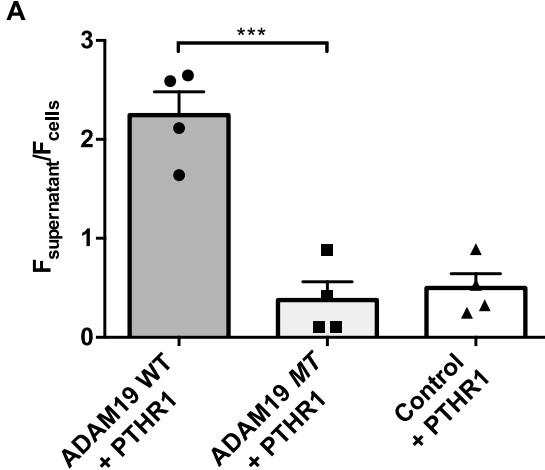

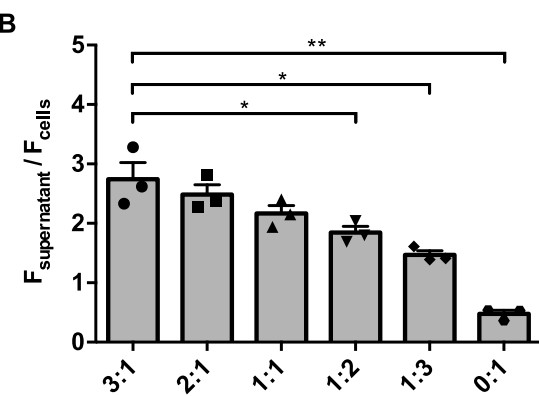

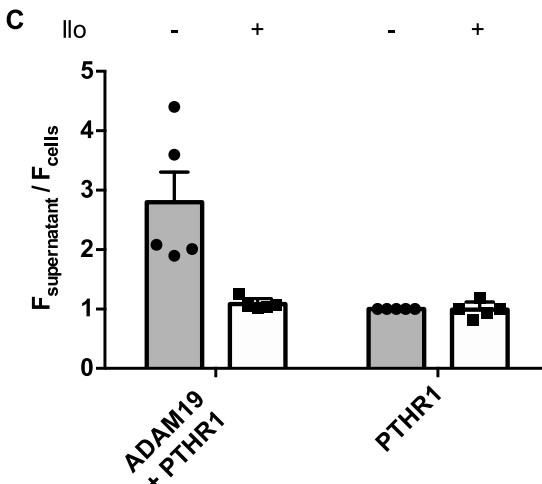

**Figure 4. Fluorescence assay for ADAM19 shedding.**
Bar-graph representation of the supernatant-cellular ratio of EYFP fluorescence, tagged at the N-terminus of the PTHR1. For fluorescence detection, HEK-293T cells were transiently transfected, once WT ADAM19 with EYFP-PTHR1 and MT ADAM19 with EYFP-PTHR1. As control, CD4 was co-transfected with EYFP-PTHR1. The data are presented as mean ± SEM; n = 4, t test, ***P < 0.001. **(A)** The data show that WT ADAM19 cleaved PTHR1, while MT ADAM19 did not (A). Different expression ratios of WT ADAM19 and EYFP-PTHR1 of 3:1, 2:1, 1:1, 1:2, 1:3 and 0:1 were analyzed. The data are presented as mean ± SEM; n = 3, t test;

Notably, here we used unmodified receptor which clearly demonstrates the proteolytic activity of ADAM19 on PTHR1 (Fig 3B).

Sheddases are membrane-bound enzymes that cleave extracellular portions of transmembrane proteins, releasing the soluble ectodomains from the cell surface. To independently confirm our findings and to test whether ADAM19 acts as a sheddase for PTHR1 on living cells, we next used an experimental approach based on fluorescence. To do so, HEK-293T cells were transiently co-transfected with EYFP-PTHR1 together with WT ADAM19, ADAM19-MT, or with CD4 as a control. By using a control that inherently does not exhibit shedding activity, we aimed to delineate the absence of shedding, thereby validating the specificity of our assay for detecting protease-mediated shedding of PTH1R, with particular focus on ADAM19. Shedding was measured by comparing the ratio of EYFP fluorescence in the cell supernatants to that of EYFP at the cell. When the EYFP-labeled PTHR1 was co-expressed with ADAM19, the majority of the fluorescence was found in the supernatant, indicating cleavage and shedding of the N-terminus of the receptor. In contrast, ADAM19-MT did not increase soluble YFP when compared to the CD4 control (Fig 4A). Notably, N-terminal receptor cleavage was dependent on the ratio between ADAM19 and PTHR1 (Fig 4B). We also examined the effect of a metalloprotease inhibitor on the cleavage behavior of ADAM19. We next incubated 100 µM Ilomastat (GM6001) in the medium for 48 h. We observed that the PTHR1 cleavage by ADAM19 was inhibited by the addition of Ilomastat (Fig 4C). A bar graph representation of the total fluorescent sum of supernatant and cell values is shown for the fluorescence assay, different expression ratio of ADAM19 and cleaved EYFP-PTHR1 ratio, and the influence of a metalloprotease inhibitor (Fig S7A–C respectively). Collectively, these data show the PTHR1 cleavage-dependency of ADAM19, and we conclude that the enzyme is a sheddase for PTHR1. We believe that the figure 4 is too large when compared to the other figures.

From the Western blot experiments, we observed that a cleavage of the PTHR1 receptor is carried out by ADAM19 and that the site is located after the amino acid position 54 (Fig 3A), which is in line with our previous findings that PTHR1 can be cleaved by metalloproteases at position Ser 61. To precisely map the cleavage site of ADAM19, we introduced various alanine substitutions at the N-terminus of EYFP-tagged receptor and measured N-terminal shedding by release of EYFP from the cell surface. To complete this task, EYFP-PTHR1 was manipulated between the amino acids 48 through 79 (referring to the native PTHR1 sequence) in segments of eight amino acid positions as shown (Fig 5A). We observed a significant decrease in soluble EYFP fluorescence between PTHR1 residues 56–71, which was most prominent for mutant 56–63A suggesting the ADAM19 cleavage site in this region of PTHR1 (Fig 5B). PTHR1 localization and internalization (trafficking) were monitored in HEK 293 cells by confocal microscopy imaging. We next

***
**P < 0.01, *P < 0.05. **(B)** The data show the PTHR1 cleavage-dependency of ADAM19 (B). Metalloproteinase inhibitor was used to analyze the activity of ADAM19 after suppressing. 1 h after transfection, cells were supplemented with DMEM or with 100 µM Illomastat (Ilo) dissolved in DMEM. As control cells were transfected with EYFP-PTHR1 without ADAM19. **(C)** The data are presented as mean ± SEM; n = 5, t test; ***P < 0.001, *P < 0.05 (C).

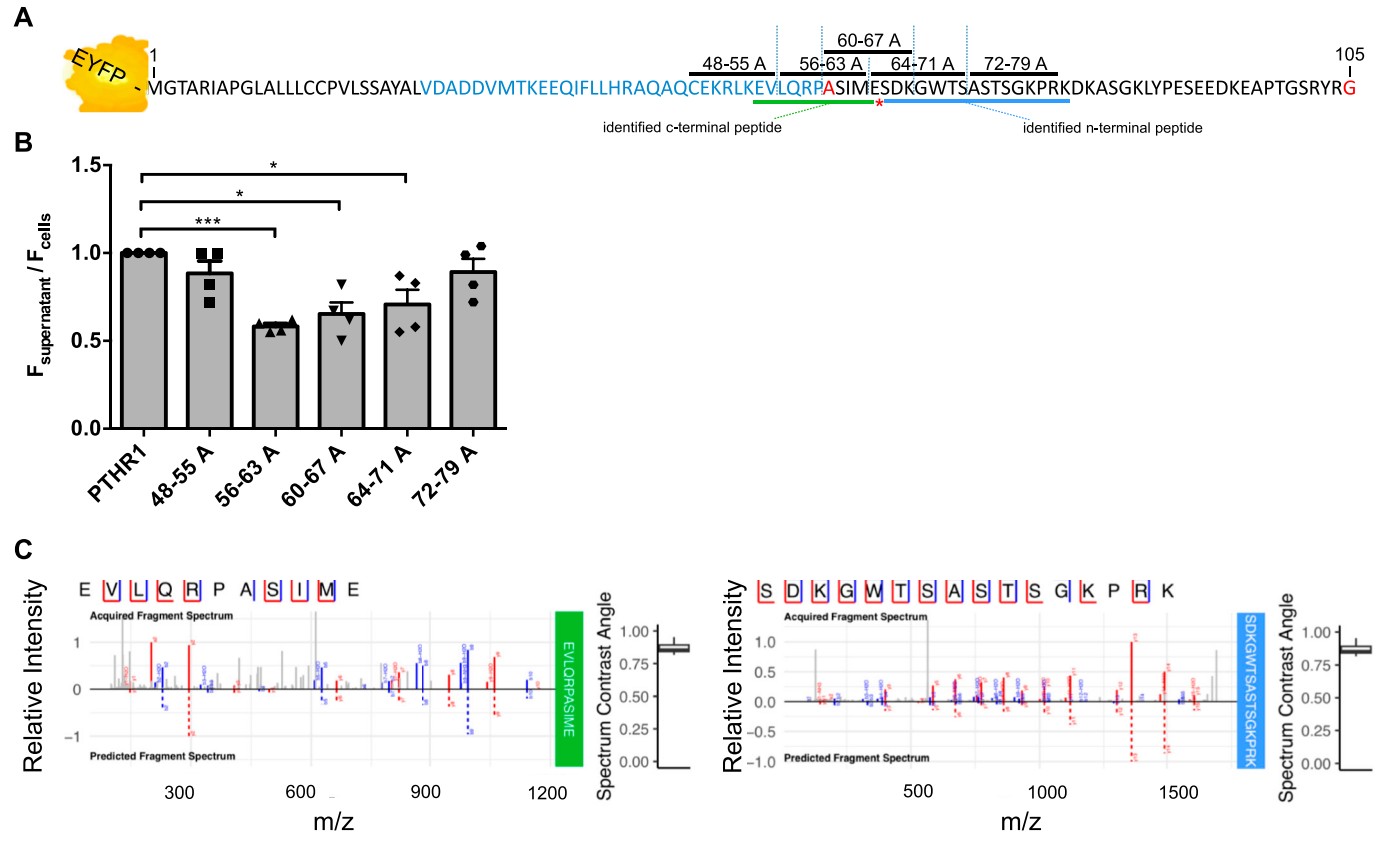

**Figure 5. Mapping the N-terminal cleavage site by alanine substitution and mass spectrometry.**
The EYFP-coupled extracellular domain of the human PTHR1 with amino acid substitutions (sequence with black line) and evaluated cleavage site (red star) is schematically shown. We manipulated amino acids 48 through 79 in segments of eight amino acids each, which were substituted by alanine. **(A)** The bars indicate the five different eight-amino acid positions which were replaced by alanine, black and blue highlighting indicates alternating exons, red highlighting indicates encoding across a splice junction (A). Bar-graph representation of the supernatant-cellular ratio of EYFP fluorescence (ordinate), with the substitution elements (abscissa). The ratio was measured at the supernatant (shedded EYFP-PTHR1 N-terminus) and the cell suspension (intact N terminus) after allowing ADAM19-mediated shedding for 48 h. Each bar represents shedding ratio of EYFP-PTHR1 with N-terminal mutations of five amino acid residues to alanine. The data are presented as mean ± SEM; n = 4, $t$ test; ***$P$ < 0.001, *$P$ < 0.05. **(B)** The mutant with alanine at 56–63 was closest to the cleavage site (B). Fragment spectra of ADAM cleavage specific peptides for (left) the c-terminal cleavage product (identifying peptide: "SDKGWTSASTSGKPRK") and (right) the n-terminal cleavage product (identifying peptide: "EVLQRPASIME"). Acquired and predicted spectra are shown in one graph for each peptide. Identified peptide-fragments are shown in blue (b-ions) and red (y-ions). **(C)** The bootstrapped spectrum similarity angle distribution is shown next to each sequence and depicts the similarity between acquired and predicted spectrum (C).

determined whether or not the alanine substitutions made the receptor behave differently in terms of internalization compared to the unmodified WT receptor. *Cellmask* was used as the membrane marker. We observed that the alanine substitutions had no visible influence on the internalization of the receptor. All manipulated amino acids between 48 and 79 revealed the same membrane localization as the WT receptor (Fig S8). To solidify these findings, we then performed mass spectrometry and could identify both c-terminal and n-terminal peptides specific for either the n-terminal or the c-terminal cleavage product of PTHR1. Based on these identifications, we can pinpoint the cleavage site to reside between amino acid position 64 and 65 (Fig 5C). This figure also seems too small.

To assess the functional implications of PTHR1 cleavage on downstream signaling, we first aimed to compare G-protein activation by PTHR1 versus the partially cleavage-protected PTHR1-56-63A mutant in the presence of ADAM19 in HEK-293T cells (Fig 6A). For this purpose, we used bioluminescence resonance energy transfer (BRET)-sensors measuring dissociation of Gα and Gβγ subunits

upon receptor-mediated G-protein activation (Schihada et al, 2021). In line with our previous findings (Klenk et al, 2022), after stimulation with PTH we observed increased $G_s$-activation and decreased $G_q$-activation for WT PTHR1 when compared to the cleavage-protected PTHR1 mutant (Fig 6B and C). After stimulation with PTHrP, likewise $G_q$-activation was decreased for WT PTHR1 compared to the PTHR1-56-63A mutant in the presence of ADAM19. However, in contrast to PTH, no differences in $G_s$-activation between cleaved and uncleaved PTHR1 were observed after PTHrP stimulation (Fig 6D and E). To corroborate these findings in a more physiological cell model, we used human osteosarcoma U2OS cells in which we measured PTHR1-mediated cAMP accumulation with the fluorescence resonance energy transfer (FRET)-based cAMP biosensor Epac-S-H187 (Klarenbeek et al, 2015), in the absence or presence of ADAM19. The U2OS is a cell line with epithelial morphology derived from an osteosarcoma. We reasoned that the U2OS cell line would exhibit a molecular background of relevant interacting proteins closely resembling those of chondrocytes, making it

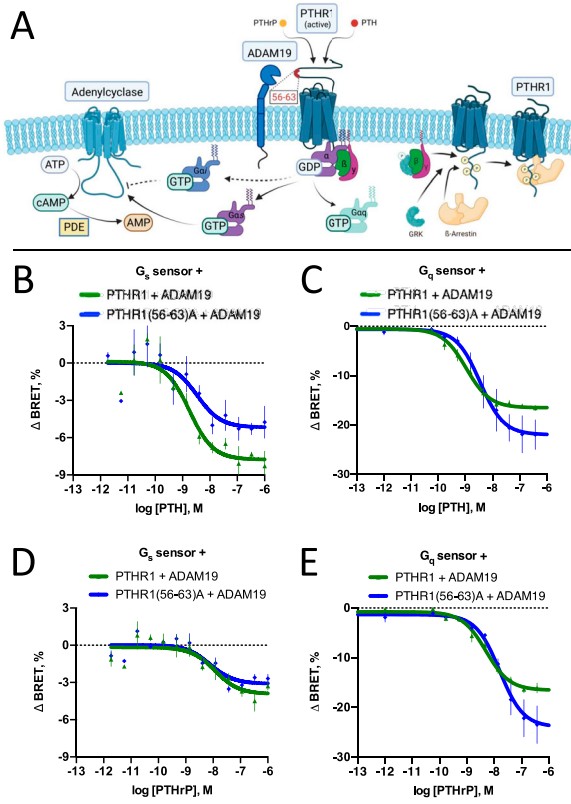

**Figure 6. Effects of ADAM19-mediated cleavage of the PTHR1 on its G-protein related signal transduction.**

**(A)** Schematic representation of cleaved PTHR1 by ADAM19 and the activation of the heterotrimeric G proteins, G$\alpha_q$ and G$\alpha_s$, effect on ß-arrestin recruitment and cAMP accumulation (A). **(B, C, D, E)** Activation of G$_s$ or G$_q$ by WT or 56–63A mutant PTHR1 in the presence of ADAM19 after stimulation with PTH (B, C) or PTHrP (D, E). Receptor variants, ADAM19 and bioluminescence resonance energy transfer (BRET) sensors for G$_s$ or G$_q$ (Schihada et al, 2021) were transiently co-expressed in HEK293T cells and changes in BRET were measured after ligand addition. Data are given as relative changes in BRET. A decrease in BRET corresponds to activation of the sensors. Data are from three independent experiments done in duplicates and represent means ± SEM. Data were fitted with a three-parameter non-linear curve fit. Top of the curve (maximum efficacy) is significantly different according to the extra sum of squares *F* test (<0.0001).

the most suitable model cell system available for our investigation. We recognize associated limitations of this cell line.

FRET measurements were performed on a single-cell basis using epifluorescence microscopy, and normalized FRET ratios were calculated as CFP/FRET (%). An increase in delta FRET, therefore, signifies an increase in cAMP throughout the cell. In response to PTH1R stimulation with both PTH1-34 and PTHrP, an increase in cAMP was detected in cells transfected with PTH1R and Epac-S-H187 alone, as well as in cells which were co-transfected with Adam19. FRET ratios were increased by 70–80% on average in both cases, visualized by representative traces (Fig 7C–F). Taken together in the responses measured in all experiments (Fig 7A and B), there were no significant differences between cells transfected with PTH1R and Epac-S-H187 and those co-transfected with ADAM19. Comprehensive control experiments indicated that YFP direct excitation values with PTHR1, ADAM19, the sensor H187 were constant across a wide range of YFP excitation values (Fig S9A–D).

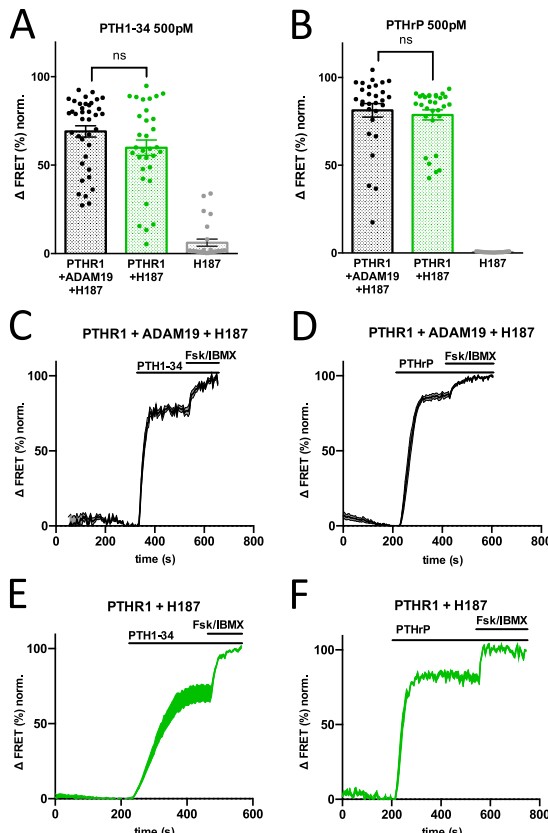

**Figure 7. PTHR1-mediated cAMP accumulation in U2OS cells in absence or presence of ADAM19.**

Levels of cAMP accumulation in single cells, measured using the Epac-S-H187 fluorescence resonance energy transfer (FRET) (Klarenbeek et al, 2015) sensor in live cell imaging. **(A, B, C, D, E, F)** Increase in ΔFRET (CFP/FRET in %) indicates an increase in cAMP (A, B, C, D, E, F). Representative traces of corrected and normalized FRET ratios. **(C, D, E, F)** U2OS cells were transfected with Epac-S-H187 and PTHR1 with and without co-transfection of ADAM, and were stimulated with 500 pM PTH1-34 (C, E) or PTHrP (D, F). The mean FRET ratio is represented by solid lines, shaded regions indicate the SEM. FRET ratios were normalized relative to the baseline (set as 0%) and the maximum stimulation by 10 $\mu$M Forskolin and 100 $\mu$M 3-iso-butyl-1 methylxanthine (set as 100%). **(C, D, E, F)** Representative traces were averaged from n = 8 (C), n = 7 (D), n = 3 (E), and n = 3 (F) cells. **(A, B)** Grouped analysis of corrected and normalized FRET ratios of all cells stimulated with 500 pM PTH1-34 (A) and PTHrP (B), from a total of n = 96 (A) and n = 76 (B) cells, measured on n = 5 (A) and n = 2 (B) experimental days. Transfection of only Epac-S-H187 was used as a control. **(A, B)** Statistics: Unpaired *t* test with Welch's correction (A) and Mann-Whitney test (B).

We then focused on PTHR1-mediated *β*-arrestin2 recruitment in human osteosarcoma U2OS and SaOS cells. Interestingly, in both cell lines PTH significantly increased *β*-arrestin translocation to the PTHR1 when ADAM19 was present (Fig 8A), similar to the results obtained for G$_s$-activation and cAMP accumulation. In contrast, after PTHrP stimulation only subtle differences in *β*-arrestin2 recruitment to PTHR1 in absence or presence of ADAM19 were observed (Fig 8B). Control expressions for cleavage and *β*-arrestin2 recruitment experiments (Fig 5B) are presented with total fluorescence (Fig S10A). The control Saos2 cell experiments (Fig S10B) with control luminescence and fluorescence data are shown (Fig S10C and D). Nonetheless, *β*-arrestin2 recruitment to PTHR1 in U2OS cells in absence or presence of ADAM19 was different (Fig 8).

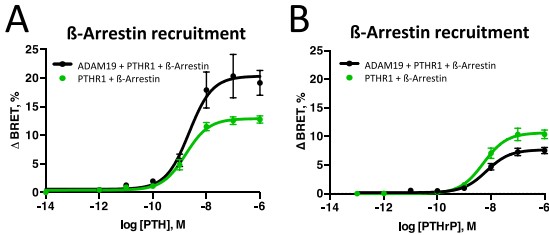

**Figure 8. β-arrestin2 recruitment to PTHR1 in U2OS cells in absence or presence of ADAM19.**
U2OS cells were transiently transfected with plasmids encoding for ADAM19, PTHR1-Nanoluc, and β-arrestin2-cpVenus (Nemec et al, 2022). **(A, B)** Arrestin2 recruitment to PTHR1 was measured after addition of PTH (A) or PTHrP (B). Data are from four experiments done in quadruple and represent means ± SEM plus individual experiments are represented as circles. Data were fitted with a three-parameter non-linear curve fit. A corresponding measurement in SaOS cells under PTH addition is shown in Fig S11B. Top of the curve (maximum efficacy) is significantly different according to the extra sum of squares F test (<0.0001).

β-arrestin recruitment was augmented in the presence of ADAM19 for PTH, while with PTHrP, β-arrestin recruitment was slightly decreased. Control expressions for cleavage and β-arrestin2 recruitment experiments for PTHrP and PTH are shown (Fig S11A and B, as well as Fig S11C and D respectively). Together, these findings suggest that PTHR1 cleavage through ADAM19 modulates differential effects on PTH-mediated receptor signaling for $G_s$, $G_q$ and β-arrestin, whereas in the case of PTHrP mainly the $G_q$ pathway is affected. Possibly, differential effects of PTH1R cleavage on PTH and PTHrP signaling could be related mechanistically. We envision altered interaction of the C-termini of PTH and PTHrP with the ECD of PTH1R (Fig S12A and B). Differences in amino acid sequences in relevant PTH and PTHrP regions could result in altered interactions within the tethered helix1 for PTH and PTHrP as shown (Fig S12C).

Finally, to bring our findings to a phenotypically more definitive (albeit speculative and preliminary) conclusion, we performed mass-spectrometry proteomics in U2OS cells. A volcano plot and heat map are presented (Fig S13A and B). We conducted a comprehensive proteomic analysis encompassing endogenous cells, cells expressing ADAM19, cells over-expressing PTHR1, and cells expressing both PTHR1 and ADAM19 concurrently. Our primary focus was to discern the distinctions between cells expressing PTHR1 (representing the disease cell model) and cells expressing both PTHR1 and ADAM19 (representing the WT model). Through application of a t test, we identified several upregulated proteins in both sets of cells. Noteworthy attention was given to genes upregulated in PTHR1-expressing cells (the disease cell model) that might hold phenotypic significance. For instance, the sugar phosphate exchanger protein 3 (SLC37A3), collagen Type I Alpha 1 Chain (COL1A1), collagen Type II Alpha 1 Chain (COL2A1), osteoclastogenesis-associated transmembrane protein 1 (OSTM1), G protein subunit gamma 7 (GNG7), and leucine rich repeat containing G protein-coupled receptor 6 (LGR6) displayed differential regulation.

In an effort to ascertain whether or not these observed protein upregulations were attributed to the overexpression of either ADAM19 or PTHR1, we compared the protein intensities across all cell lines. Our findings indicated that the noted proteins that are particularly intriguing, exhibited diminished abundance in cells

solely expressing ADAM19. This observation suggests that the effect is more likely attributed to the presence of PTHR1 rather than to ADAM19.

## Discussion

We showed that ADAM19 is a metalloproteinase that cleaves the N-terminus of the PTH receptor. Earlier work from our group demonstrated that the ECD of PTHR1 is subject to metalloproteinase cleavage (Klenk et al, 2010a). Recently, our group localized the metalloproteinase cleavage site to the first loop of the PTHR1 ECD (Klenk et al, 2022). The cleaved receptor exhibited enhanced signaling to $G_s$ and decreased activation of $G_q$ compared to WT PTHR1. The findings reported here verified ADAM19 as a responsible metalloprotease capable of such cleavage. We identified the ADAM19 locus by means of genetic linkage in a small pedigree, solely exhibiting autosomal-dominant BDE. Genome sequencing subsequently revealed a truncated ADAM19 allele that was identified in all affected family members and in none of nonaffected persons. Since our earlier studies had shown that PTHrP is responsible for BDE, we tested the hypothesis that ADAM19 interacts with the PTHrP-PTHR1 signaling pathway. We found that ADAM19 is capable of PTHR1 cleavage and showed that the truncated ADAM19 was unable to cleave PTHR1. We used alanine substitutions to map the N-terminal cleavage site of PTHR1. Our mass-spectrometry data on the cleavage site provide convincing evidence that ADAM19 is indeed a *sheddase* for PTHR1. Finally, we examined the effects of PTHR1 cleavage on downstream signaling and compared G-protein activation by the cleaved versus the noncleaved PTHR1. We found that protection from the ADAM19 cleavage of its N-terminal ECD shifts signaling of the PTHR1 between the major downstream pathways. Noncleaved PTH1R exhibited increased activation of $G_q$ and decreased activation of $G_s$ in response to PTH stimulation when compared to cleaved PTH1R. Likewise, PTH-mediated β-arrestin recruitment to PTH1R was reduced in the absence of ADAM19. In contrast, cleavage-mediated effects on receptor signaling were more directed after stimulation with PTHrP. Cleavage protection of PTH1R resulted in increased $G_q$ signaling whereas only subtle changes in $G_s$ activation and β-arrestin recruitment were found. Although PTH and PTHrP are both full agonists for PTH1R, their binding kinetics and their selectivity for distinct receptor conformations differ (Hattersley et al, 2016). While the N-termini of both peptides have high sequence similarity and largely make comparable molecular contacts to the transmembrane domain of PTH1R, the C-terminal regions of the peptide ligands are less conserved resulting in marked differences in their interaction with the ECD (Pioszak et al, 2009; Ehrenmann et al, 2018; Kobayashi et al, 2022) (Fig S12). Importantly, several of these non-conserved residues make contacts with the N-terminal α1 helix of the ECD and have been shown to strongly influence the differences in binding kinetics and signaling properties between both ligands (Mannstadt et al, 1998; Mann et al, 2008; Pioszak et al, 2009). Proteolytic cleavage of the ECD at position 64 would likely increase the flexibility of the N-terminal α1 helix, which is located proximal to the cleavage site, but remains tethered to the distal part of the ECD through a disulfide bond between C64 and C117 (Klenk et al, 2010a) (Fig S12). Given the

ligand-specific contacts at the proximal end of α1 helix, it is well conceivable that increased flexibility of the binding pocket after receptor cleavage may result in the observed ligand-dependent changes in receptor signaling. This hypothesis is further corroborated by the notion that in other congenital growth disorders, underlying mutations in the ECD of PTH1R are found. Homozygous mutation of E35, which is located at the N-terminal end of α1 helix and which makes contacts to the variant residues 16 and 19 in PTH/PTHrP, is found in the rare Eiken syndrome characterized by delayed bone mineralization and epiphyseal dysplasia (Moirangthem et al, 2018). Notably, this mutation leads to distinct changes in signaling response to PTH and PTHrP similar to what has been observed in the present study (Moirangthem et al, 2018). Altered Signaling and Desensitization Responses in PTH1R Mutants Associated with Eiken Syndrome have been described (Portales-Castillo et al, 2023). Together, these findings underscore the importance of extracellular regions of GPCR for fine-tuning and adjusting receptor activity and signaling selectivity.

Our study potentially adds human phenotypes to metalloprotease cleavage of PTHR1. In earlier studies, we identified PTHrP as responsible for BDE (Maass et al, 2010, 2012, 2015, 2018). Mutated *PTHLH* encoding PTHrP has also been shown to produce the BDE phenotype (Klopocki et al, 2010). Since PTHrP signals by means of PTHR1, we were interested in what interactions ADAM19 might have with bone and cartilage development (Suva & Friedman, 2020). Chesneau and colleagues investigated the catalytic properties of ADAM19 (Chesneau et al, 2003). They found that the enzyme splits TNF-α cleavage sites, osteoprotegerin ligand, and Kit ligand. The authors identified a role for ADAM19 in regulation of the shedding process. The ADAM19 targets were generously expressed in osteoblasts. Amongst other metalloproteinases, ADAM19 participated in cartilage development. A role for the enzyme was identified in studies of microenvironmental changes and mesenchymal stem-cell differentiation towards chondrocytes (Djouad et al, 2007).

In higher vertebrates, cephalic neural crest cells form the craniofacial skeleton by differentiating into chondrocytes and osteoblasts. Arai et al recently identified a ADAM19 as a position-specific fate regulator of neural crest cells (Arai et al, 2019). ADAM19 mediated the cleavage of bone morphogenic protein (BMP) receptor type I (Alk2). The group also presented evidence that the sex determining region Y-box 9 (Sox9) cascade was suppressed presumably related to altered Alk2 processing. ADAM19 has also recently been identified as a microRNA target. Kong et al found that synovial mesenchymal stem cell-derived miR-320c targeted ADAM19 directly, thereby enhancing chondrogenesis (Kong et al, 2022).

In BDE, metacarpals and metatarsals are selectively shortened. Chondrocyte hypertrophy is accompanied by increased organelle synthesis and rapid intracellular water uptake, which serve as the major drivers of longitudinal bone growth. This process is delicately regulated by major signaling pathways and their target genes, including growth hormone, insulin growth factor-1, Indian hedgehog (IHH), PTHrP, BMPs, Sox9, runt-related transcription factors (Runx) and fibroblast growth factor receptors (Hallett et al, 2021).

PTHrP was identified as causing the hypercalcemic paraneoplastic syndrome. Its receptor and the encoding gene were subsequently identified (Clemens et al, 2001). *PTHLH* is expressed in the distal ends of cartilage anlagen and in the perichondrium of developing bones. PTHrP diffuses away from its site of production and binds to PTHR1 located on prehypertrophic chondrocytes. PTHrP binding results in PTHR1 and later $G\alpha_s$ activation, which in turn keeps chondrocytes proliferating through suppression of the cyclin-cdk inhibitor p57, thereby increasing the pool of proliferating nondifferentiated chondrocytes (Kronenberg, 2006). PTHrP synthesis is controlled by IHH (St-Jacques et al, 1999). The pathologic, molecular, and clinical correlates, including signaling involved in disorders of G protein-mediated cAMP activation have been reviewed (Cohen, 2006).

Presumably, heterozygosity for ADAM19 in our affected BDE patients reduced their ADAM19 activity and promoted a relative increased activation of $G_q$ and decreased activation of $G_s$, perhaps at a critical stage in cartilage development. PTHR1 mutations resulting in receptor loss of function have been associated with endochondral developmental defects earlier (Couvineau et al, 2008). The receptor's signaling responses to PTH and PTHrP is currently being elucidated with high resolution near-atomic structural techniques (Sutkeviciute et al, 2019). Such efforts could contribute to further mechanistic explanations of our BDE syndrome. We investigated plasma membrane-related $G_s$ and $G_q$ effects; however, endosomal cAMP responses possibly also play a role. For instance, PTHR1-related $G_s$ and $G_{q/11}$ activation has been shown to mediate endosomal cAMP responses (Ferrandon et al, 2009; Jean-Alphonse et al, 2017; White et al, 2020).

The observed differences in PTH and PTHrP effects on PTH1R-related $G_s$ activation could potentially stem from variations in the execution-time points of the assays (different days and different passage of the cells). Our observations suggest that the dynamic range of the $G_s$ BRET assay exhibits greater variability compared to the $G_q$ BRET assay that was more stable and had lower day-to-day variability. Alternatively, the observed differences could arise from the specific timepoint chosen for generating the concentration-response curve, as PTHrP-mediated $G_s$ activation reaches saturation at a distinct point compared to PTH-mediated activation.

The FRET experiments, which measure cAMP accumulation, inherently evaluate processes further downstream in the signaling cascade, introducing the possibility of other factors masking the observed effects closer to the receptor, such as enhanced β-arrestin interactions (Fig 8). Additionally, the transient transfection of ADAM19 and thus incomplete cleavage (Fig 5B), different cell line used (HEK293 versus U2OS), and different heterologous expression protocol (transfection versus transduction) could potentially obscure downstream effects of enhanced $G_s$ signaling. With these caveats in mind, we nonetheless suggest that our genetic findings support the idea that mutated ADAM19 is associated with our clinical phenotypes.

Together, IHH and PTHrP form a feedback loop regulating chondrocyte hypertrophic differentiation and thereby endochondral bone development (Vortkamp et al, 1996). The effects of PTHrP on chondrocyte differentiation are also mediated by means of the Sox9 phosphorylation, a transcription factor important for chondrocyte differentiation, and by suppressing Runx2. We identified Sox9 participation in BDE earlier (Maass et al, 2018). Runx2 is a transcription factor essential for osteoblast differentiation.

Independent of our findings here, ADAM19 is implicated in bone development, BMP, and Sox9 signaling.

Our findings that ADAM19 cleaves PTHR1 underscore the relevance of these observations. The BDE syndromes we encountered earlier and the family we describe here had no abnormalities in calcium or phosphorus homeostasis. Furthermore, their PTH levels, when measured were not perturbed. We have insufficient data on PTHrP concentrations in our patients; however, PTHrP exerts paracrine functions and is not a circulating hormone aside from overproduction by tumors in paraneoplastic syndromes.

We found an effect of ADAM19 cleavage on β-arrestin recruitment. The finding was driven by a report on PTH1R-modification by such enzymes (Klenk et al, 2010a). GPCRs activate heterotrimeric G proteins. In order to turn off a response, or to adapt to a persistent stimulus, active receptors must be desensitized. The first step in desensitization is phosphorylation of the receptor by G protein-coupled receptor kinase. Arrestin binding to the receptor blocks further G protein-mediated signaling and targets receptors for internalization, and redirects signaling to alternative G protein-independent pathways, such as β-arrestin signaling. The importance of arrestins in general and in PTHR1-related signaling in particular is well established (Lohse et al, 1990; Gesty-Palmer et al, 2006; Ferrari & Bouxsein, 2009; Klenk et al, 2010b). β-arrestin is also important in modulating opposing effects of PTHR1 in calcium-sensing receptor signaling (Gorvin et al, 2018). Moreover, receptor activity-modifying protein 2 was shown to modulate PTHR1-stimulated amounts of ß-arrestin2 recruitment and downstream signaling (Nemec et al, 2022). GPCR signaling by PTHrP and fragments has been investigated, including effects on β-arrestin2 (Pena et al, 2022).

The lack of ADAM19 cleavage effects on cAMP responses to PTH in U2OS cells warrants explanations. Increased β-arrestin coupling could counteract the observed rise in $G_s$ activation and mask increased cAMP accumulation. Additionally, the choice of cell type (HEK293 versus U2OS) and protocols for heterologous expression could impact the outcomes, as the stoichiometry and expression levels of interacting proteins can vary. The cAMP accumulation was evaluated in U2OS cells, where we opted for transduction over transfection to enhance assay performance.

Recently, Chu and colleagues reported that matrix metalloprotein-14 (MMP14) cleaves PTH1R in the chondrocyte-derived osteoblast lineage, curbing signaling intensity for proper bone anabolism (Chu et al, 2023). Their study identifies a novel paradigm of MMP14 activity-mediated modulation of PTH signaling in the osteoblast lineage, contributing new insights into bone metabolism. Their findings suggested that MMP14 dampens PTH signaling. Our investigation was rather focused on PTHrP. Nonetheless, the findings underscore the role of regulatory enzymes in PTHR1 signaling. Conceivably, other such enzymes influence PTHR1 activity.

Modulation of PTH1R signaling by a large ECD binding antibody results in inhibition of β-arrestin 2 coupling (Sarkar et al, 2019). Others (Chu et al, 2023) and we (Klenk et al, 2010a) have shown earlier that PTH1R-ECD can also be subject to cleavage by proteases other than ADAM19. Indeed, we observe a minor fraction of processed PTH1R that may be cleaved by other metalloproteases

endogenously expressed in HEK293 cells, the cell line that was employed for the experiments.

The "warts, hypogammaglobulinemia, infections, and myelokathexis" syndrome is an inherited immune disorder caused by an autosomal dominant heterozygous mutation in CXCR4. Kumar et al (2023) recently found reduced GPCR signaling despite impaired internalization and β-arrestin recruitment in patients carrying a CXCR4Leu317fsX3 mutation causing warts, hypogammaglobulinemia, infections, and myelokathexis syndrome. Our data show that in the presence of ADAM19, lesser amount of β-arrestin2 are recruited with PTHrP, compared to the effects of PTH. We suggest that differential β-arrestin2-related effects in response to ligands could have contributed to our findings. LGR6 is a Wnt-associated adult stem-cell marker and is required for osteogenesis and fracture healing (Doherty et al, 2023).

PTHrP signaling could also be of relevance above-and-beyond development. PTHrP is abundant in osteoarthritic cartilage and a PTHrP isoform has been implicated in faulty mineralization of osteoarthritic cartilage (Terkeltaub et al, 1998). PTHrP signaling has also been identified in tumor metastases (Ponzetti & Rucci, 2020). PTHrP expression in breast cancer is enriched in bone metastases compared to primary tumors. The protein exerts a cytokine-like function in tumor microdomains, where it serves a paracrine function to promote breast-cancer metastases (Johnson et al, 2018). Independent of PTHrP, ADAM19 is also implicated in the oncogenic properties of breast cancer. The oncohistone, H2BE76K upregulated ADAM19, which in turn participated in enhanced tumor colony formation that could be inhibited by knock-down (Kang et al, 2021).

Finally, we attempted to identify possible molecular candidate genes that could mechanistically contribute to our findings. SLC37A3 is a transporter complex responsible for the cytosolic entry of nitrogen-containing bisphosphonates, important to bone metabolism (Yu et al, 2018). The collagen-encoding genes are obvious fits. The osteopetrosis-associated transmembrane protein 1 (OSTM1) causes craniofacial and dental abnormalities (Ma et al, 2023). These genes could serve as guidance parameters for future studies.

Our data have limitations. The evidence that our ADAM19 mutation causes BDE is not circumstantial; we can offer no direct proof. Zhou et al generated ADAM19 gene-deleted mice (Zhou et al, 2004). Homozygous mice exhibited severe cardiac defects. Oddly, although the mice were smaller, ADAM19 did not appear essential for bone growth and skeletal defects. ADAM19 was expressed in proliferating chondrocytes distal to IHH expression. Whole mounts of paws exhibited similar bone development of ADAM19−/−, ADAM19−/+, and ADAM19+/+ mice. Our patients are heterozygous and exhibited no cardiac defects.

Nevertheless, "mouse is not man;" for instance, the impressive effect of ADAM−/− on cardiac defects has not translated into documenting the effects of ADAM19 on human congenital heart disease. Furthermore, the facts that ADAM19 is expressed adjacent to IHH in the paws of mice during development and that ADAM19 cleaves the signaling receptor of PTHrP, the known driver of BDE, is compelling. We suggest that ADAM19 is a sheddase for PTHR1 and is implicated in regulating the diverse functions of this important GPCR.

# Materials and Methods

### Whole-genome mapping of the linkage regions in the family

University of the Witwatersrand Ethical Committee approved the study and written, informed consent from all participants was obtained. Local approval was provided by the ethics committee (Charité-Universitätsmedizin Berlin, Ethikausschuss 1; EA3/009/07) who granted the analyses. Whole-genomic DNA was extracted from EDTA-blood using standard procedures. We performed a genome-wide scan in all participating family members (six affected and two non-affected) with Affymetrix Human Genome-Wide SNP 6 (>930 k SNPs) array according to the assay. Genotype calling was performed with the Affymetrix Genotyping Console v2.1 using the Birdseed-v2 algorithm. Quality control and data handling were managed by ALOHOMORA (Ruschendorf & Nurnberg, 2005). Genotyping errors were analyzed with Pedcheck (O'Connell & Weeks, 1998) and unlikely genotypes were identified with MERLIN (Abecasis et al, 2002) and removed from the dataset. Linkage analyses were then performed with 109,096 SNPs using MERLIN (Abecasis et al, 2002), with a fully penetrant autosomal dominant mode of inheritance with no phenocopy and with allele frequencies from a European population. In Linkage regions a maximal LOD score of 1,505 could be detected for this family.

### Verification of Linkage regions with microsatellite markers

The five longest Linkage regions (>2 MB) were genotyped with 11, 8, 6, 9 and 10 microsatellite markers. After individual PCR amplification, products were pooled and size-fractionated by electrophoresis on MegaBase 1,000 (Amersham Pharmacia) or ABI 3100 (Applied Biosystems) DNA capillary sequencers. Individual markers were completely typed on one genotyping station. We determined allele sizes using the GeneticProfiler (Amersham Pharmacia) or Genescan 2.1.1. and Genotyper V2 software (Applied Biosystems). All marker genotypes were checked for Mendelian inheritance using the PedCheck software as aforementioned Pedcheck (O'Connell & Weeks, 1998). Linkage analysis with microsatellite markers verified the maximal LOD score of 1.5 in the five longest linkage regions.

### Whole-genome sequencing and Sanger sequencing

Whole-genome sequencing of two affected patients and one control from a family was performed (Complete Genomics) (Drmanac et al, 2010). After bioinformatics analysis, data were further analyzed with CGA tools. Numerous insertions, deletions or substitution events within the linkage region, were found. After excluding annotated SNPs, we selected heterozygous variants that were not detectable in the unaffected control or in the hg19 genome assembly. Among the Complete Genomics–annotated variants in coding genes within the linkage interval, a mutation in *ADAM19* was detected in the two affected patients. The mutation was further analyzed in all available family members (five affected and two unaffected individuals) with Sanger resequencing. After identification of the same mutation by Sanger resequencing in all affected family members, alleles were subcloned in order to precisely identify the mutation sites. After PCR amplification with the forward primer (5'- aatgacatcttccctgcccc) and the reverse primer (5'-gctggcttgcacacattctc) amplicons were sequenced using BigDye Terminator Cycle Sequencing Kit v1.1 (Applied Biosystems). Analysis was performed on a 3130xl Genetic Analyzer (Applied Biosystems) using Gene Mapper Software Version 4.0. SeqMan Software (Lasergene, Version 15.0; DNAStar) was used to evaluate the tracks.

### Expression plasmids

The following vectors were used for transfection and mutagenesis experiments: The plasmid ADAM19 and PTHR1 was purchased from OriGene (#RC222510 and #RC212841). All mutations were introduced into the full-length cDNA clone by in vitro mutagenesis according to the manufacturers' protocols (QuickChange II XL Site-Directed Mutagenesis kit, Ambion Technologies or Q5 Site-Directed Mutagenesis Kit, New England Biolabs). $G\alpha_s$ and $G\alpha_q$ $\beta$-Arrestin2 protein BRET biosensors were a gift from Hannes Schihada and were described previously (Schihada et al, 2021).

### Cell culture

HEK-293T (ECACC #96121229; Sigma-Aldrich) cells were routinely cultured in DMEM supplemented with 10% (vol/vol) FBS (Biochrom AG), 1% L-glutamine (PAN Biotech), penicillin (100 U/ml), and streptomycin (100 $\mu$g/ml) at 37°C and 5% $CO_2$. SaOS (#HTB-85; ATCC) and U2OS (# HTB-96; ATCC) cells were grown in DMEM supplemented with 10% (vol/vol) FBS, 1% L-glutamine, penicillin (100 $\mu$g/ml) and streptomycin (100 $\mu$g/ml).

### Immunoprecipitation of FLAG-tagged ADAM19

HEK-293T cells were seeded in six-well cell culture plates at a density of 500,000 cells per well 24 h prior to transfection. Cells were transfected with 30 $\mu$l PEI (1 mg/ml; linear polyethylenimine 25,000, Polysciences, Inc.) and 1–2 $\mu$g ADAM-FLAG constructs (WT and mutated) and cultured for another 48 h. Cells were scraped in RIPA buffer (150 mM NaCl, 50 ml Tris–HCL pH 7.8 in PBS, 0.5% sodium deoxycholate, 0.1% SDS) supplemented with protease and phosphatase inhibitors (Complete and PhosSTOP, Roche Diagnostics) and lysed by use of 26G needles (Sterican, Braun). Lysates were cleared by centrifugation (21,250*g*, 15 min, 4°C) and protein concentration was determined with Bradford reagent (Thermo Fisher Scientific). Anti-Flag M2 Magnetic Beads (Sigma-Aldrich) were washed with TBS, supplemented with protease and phosphatase inhibitors. The beads (50 $\mu$l) were added to 0.5 mg of protein and incubated overnight at 4°C on a rotating device. The supernatant was denatured with 4× Lämmli sample buffer (50 mM Tris–HCl, pH 6.8, 4% glycerol, 1.6% SDS with 4% $\beta$-mercapto-ethanol), and the beads were washed with cold TBS, supplemented with protease and phosphatase inhibitors (Complete and PhosSTOP, Roche Diagnostics). 4× Lämmli sample buffer was added and the samples were boiled at 95°C for 5 min.

## Receptor de-glycosylation

HEK-293T cells were seeded onto 12-well plates and transfected with plasmids encoding human PTHR1, ADAM-19 using TransIT-293 (Mirus) according to the manufacturer's instructions. Protease inhibitors were added to the medium 24 h after transfection as indicated. 36 h after transfection, cells were washed with PBS and lysed in ice-cold radioimmune precipitation assay buffer (50 mM Tris–HCl, pH 7.5, 150 mM NaCl, 1% [wt/vol] Nonidet P-40, 0.1% [wt/vol] SDS, 0.5% [wt/vol] sodium deoxycholate, 5 mM EDTA) supplemented with a mix of protease inhibitors (5 $\mu$M leupeptin, 5 $\mu$M pepstatin-A, 1 mM 4-(2-aminoethyl) benzenesulfonyl fluoride). The lysates were cleared by centrifugation at 20,000$g$ for 30 min at 4°C. For deglycosylation, lysates were incubated with 40 mM dethiothreitol and 0.5% SDS for 90 min at 25°C. Thereafter, 1 mg/ml PNGase F was added, and samples were incubated for 16 h at 4°C.

## Western blotting

Proteins were separated by 8% or 10% SDS–PAGE and transferred onto PVDF membranes (0.45 mM, #T830.1; Carl Roth). The membranes were blocked in 5% skimmed milk powder (Sigma-Aldrich) or in casein solution (Sigma-Aldrich) for 1 h at RT. Primary antibodies were incubated overnight at 4°C in 5% skimmed milk and 0.1% Tween. Membranes were washed 3× in TBS/0.05% Tween and incubated with secondary antibody for 1 h at RT. Membranes were washed 3× in TBS/0.05% Tween. Proteins were detected either after short incubation in Immobilon Western ECL substrate (#WBKLS0500; Millipore) on an Odyssey FC device (Li-cor Biosciences) or on an Odyssey CLx scanner (Li-cor Biosciences). The following antibodies were used: ADAM19$_{74-123}$ (#NBP1-69367; Novus), ADAM19$_{218-267}$ (#ab104800; Abcam), FLAG (#TA50011-100; Origin), PTHR1$_{4-54}$ (#ABIN2776779; Antibodies-Online.de), PTHR1$_{52-86}$ (#ABIN5708269; Antibodies-Online.de), FLAG (#TA50011-100; Origin), HA (#137838; Abcam), polyclonal PTHR1$_{573-593}$ (Lupp et al, 2010), anti-mouse IgG-HRP (#P0260; Dako), anti-rabbit IrDye680 (#926-32223; Li-cor Biosciences).

## Whole-mount RNA in situ hybridization

In situ hybridizations were carried out using digoxygenin-UTP-labeled sense and antisense riboprobes for mouse *Adam19* (coding sequence probe, chr11:46,127,331-46,136,268 bp; 3'UTR probe, chr11:46,143,961-46,144,454 bp; UCSC Genome Browser mm10) according to standard protocols used earlier (Maass et al, 2015).

## Immunofluorescence and confocal microscopy

Transiently transfected HEK-293T cells were grown on glass coverslips. After one PBS wash, cells were fixed with fresh 4% paraformaldehyde for 10 min at RT and permeabilized with 80% methanol for 20 min at −20°C. After blocking for 1 h with 2% BSA in PBS and incubation with primary FLAG antibody (at the dilution of 1:200) staining was accomplished with Alexa Fluor 488-coupled (#A110291; Thermo Fisher Scientific) secondary antibodies. Cell nuclei were counterstained with DAPI (#D1306; Invitrogen). 1:10,000 diluted 0.1 $\mu$g/ml of DAPI was always added into the mixture of secondary antibodies. After washing step, subsequently, the slides were mounted with Aqua-Poly/Mount medium (#18606; Polysciences) and stored at 2–8°C in a dark box. Microscopic images were taken with an inverted TCS SP5 tandem confocal microscope (Leica Microsystems). Image processing was done using ImageJ/Fiji software.

## ADAM19 shedding assay

For fluorescence-based shedding experiments, HEK-293T cells were seeded in six-well cell culture plates at a density of 500,000 cells per well 24 h prior to transfection. Next day, cells in each well were transfected with 0.4 $\mu$g EYFP-PTHR1 plasmid and 0.8 $\mu$g ADAM19 or empty pcDNA3 vector plasmids. For the experiments in which the different ADAM:EYFP-PTHR1 ratios were examined, cells were transfected with a constant 0.3 $\mu$g EYFP-PTHR1 construct alongside with varying amounts of ADAM19 and pcDNA constructs to maintain the total DNA amount of 1.2 $\mu$g. 1 h after transfection, cells were supplemented with DMEM or with 100 $\mu$M ilomastat dissolved in DMEM. 47 h after treatment, culture medium from each well was collected and centrifuged at 5,000 rpm for 5 min (Eppendorf Centrifuge 5425/5425 R). Cells remaining on six-well plates were washed once with PBS and then were suspended in 2.2 ml DMEM. Collected culture medium and cells were distributed in black-walled, black-bottomed 96-well plates (200 $\mu$l per well). 96-well plate fluorescence measurements were performed using a Biotek Neo2 multi-well plate reader. EYFP intensity was measured using high lamp power settings at an excitation wavelength of 500/18 nm. Fluorescence was detected using a monochromator set to 539/20 nm for emission and a photomultiplier tube set to 120 V gain. To calculate and quantify the ADAM19-mediated shedding of PTHR1 N-terminus, background fluorescence value obtained from pure DMEM was subtracted from fluorescence values obtained from cell culture medium ($F_{supernatant}$) and cell suspensions ($F_{cell}$), and then the ratio of background-corrected fluorescence values was calculated: Ratio = $F_{supernatant}/F_{cell}$. Calculated ratios were then normalized by the ratio of the control condition (in which the cells were transfected with EYFP-PTHR1 + pcDNA3).

## BRET-based Gs, Gq activation and β-arrestin2 recruitment protein assay

U2OS, Saos2 and HEK-293T cells were transfected with $G_s$ or $G_q$ protein BRET biosensors (Che et al, 2010), PTHR1 and ADAM19 at a ratio of 1:1:2. For β-arrestin2 BRET recruitment assay, PTHR1-Nanoluc was used with β-arrestin2-cpVenus and ADAM19 at a ratio of 1:1:2 (Nemec et al, 2022). Combinations were transfected with Effectene (QIAGEN) or NEON electroporation (Thermo Fisher Scientific) according to the manufacturer's protocol with a total of 1–2.4 $\mu$g of cDNA. After 24 h cells were transferred into a PDL-precoated a white-wall, white-bottomed 96-well microtiter plate, at a density of 60,000 cells/well. 24 h after the reseeding, the medium was removed, and cells were washed once with PBS with 0.1% BSA and incubated with 90 $\mu$l of a 1:1,000 (vol:vol) stock solution of furimazine in PBS buffer. 5 min later, basal reads were recorded for 4 min and subsequently, 10 $\mu$l of 10-fold ligand solution or FRET buffer was applied to each well and the stimulated reads were further recorded. Measurements were performed at 37°C using a

Synergy Neo2 Plate Reader with the NanoBRET filter set, integration time per data point was set to 0.3 s and gain to 110/140. Cells were excited at 510/20 nm and fluorescence emission was recorded at 560/20 nm for quantification of expression level (Nemec et al, 2022).

### FRET-based cAMP accumulation assay

For cAMP accumulation assays, U2OS cells were seeded on Poly-D-Lysine-coated 24 mm glass cover slips in 6-well plates at a density of $1.0 \times 10^6$ cells/well and transfected with 1–2.4 $\mu$g cDNA per cover slip using Effectene (QIAGEN) or NEON electroporation (Thermo Fisher Scientific) according to the manufacturer's protocol. cDNA was transfected with the EPAC-S-H187 FRET-biosensor (Klarenbeek et al, 2015), PTHR1-wt and ADAM19 at a ratio of 1:1:2. Fluorescence microscopy experiments were performed 48 h after transfection. For single-cell FRET imaging experiments, transfected cells on coverslips were transferred to imaging chambers (Attoflur; Thermo Fisher Scientific) and washed once and maintained in FRET imaging buffer (144 mM NaCl, 5.4 mM KCl, 2 mM CaCl$_2$ [Carl Roth GmbH & Co. KG], 1 mM MgCl 2 [AppliChem], 10 mM HEPES [Sigma-Aldrich Chemie GmbH]; pH = 7.3). For the experiment, the cells remained at RT throughout the experiment.

FRET measurements were performed using an epifluorescence microscope (Leica DMi8 inverted microscope; Leica Microsystems) with an oil immersion objective (HC PL APO 40x/1.30; Leica Microsystems), a dichroic T505lpxr beam splitter (Visitron Systems), a xenon lamp (75 W, 5.7 A; Hamamatsu Photonics) coupled to a high-speed polychromator (VisiChrome, Visitron Systems), and a Photometrics Prime 95B CMOS camera (Visitron systems) with an Optosplit II dual emission image splitter (Cairn research, Edinburgh, Scotland, UK) containing CFP 470/24 and YFP 535/30 emission filters (Chroma Technology).

After selecting suitable cells at the microscope, regions of interest (single cells and a cell-free background region) were chosen and recording parameters were set using the Visiview 4.0 imaging software (Visitron Systems). Every 5 s, images of the donor and acceptor emission channels were recorded following donor excitation at 436 nm for 100 ms. Once the emission ratio traces reached a baseline, the PTHR1 ligand (500 pM PTH1-34 BACHEM #4033364 or PTHrP, BACHEM # 4017147) was applied to the coverslip using a pipette. To achieve the maximum cAMP response, 10 $\mu$M Forskolin (Sigma-Aldrich) and 100 $\mu$M 3-iso-butyl-1 methylxanthine (Sigma-Aldrich) were applied at the end of the experiment. After every experiment, direct YFP excitation at 505 nm (emission: 560 nm) was recorded as a unit of measurement for the expression level of the sensor.

For data analysis, raw emission intensities from CFP and YFP channels were exported and corrected for background (by subtracting the emission intensities of a cell free region) and bleed-through of the donor fluorescence into the acceptor channel (Borner et al, 2011). FRET ratios (CFP/FRET in %) were calculated and normalized to baseline average of 10 data points before ligand addition, set to 0% and Forskolin/3-iso-butyl-1 methylxanthine (max. cAMP response, set to 100%).

Data analysis and statistical tests were performed using Prism software 9.0 (GraphPad Software). Normal distribution of the data points was evaluated using a D'Agostino-Pearson normality test. To compare two datasets with normal distribution, a t test with Welch's correction was used and the confidence interval was set to 95% (P-value = 0.05). For non-normal distributions, a Mann Whitney test was used. Data are represented as mean ± SEM.

### RNA extraction, cDNA synthesis and real-time quantitative PCR

Extraction of total RNA was performed according to the Trizol manufacturer's protocol (Invitrogen). For tissue samples 1 $\mu$g of total RNA was used for cDNA synthesis. cDNA synthesis was performed using SuperScript II (Invitrogen) and hexanucleotide primers according to manufacturer's protocol. The real-time quantitative PCR assays were performed using the 7,500 Real-Time PCR System (Applied Biosystems). Amplifications were carried out in 25 $\mu$l reaction solutions containing 12.5 $\mu$l 2× SyberGreen Mix Separate-ROX (Roche), 2.5 $\mu$l first-stranded cDNA (diluted 1:10), 333 mM of each specific primer. PCR conditions were 95°C for 2 min followed by 40 cycles of 95°C for 5 s and 60°C for 60 s. To check reproducibility, the assay was performed with technical triplicates for each biological sample. PCR efficiency values (E) were calculated for ADAM19 gene from the given slope after running standard curves and following the formula $E = (10^{(-1/slope)} - 1) \times 100$.

### Mass spectrometry

**For cleavage evaluation** HEK-293 cells were seeded at a density of $2.5 \times 10^6$ cells per 6 cm dish 24 h prior to transfection. Next day, cells in each well were transfected with 0.7 $\mu$g EYFP-PTHR1 and 1.4 $\mu$g ADAM19 or empty pcDNA3 vector plasmids. Combinations were transfected with Effectene (QIAGEN) according to the manufacturer's protocol with a total of 2.8 $\mu$g of cDNA and cultured for another 48 h. Culture medium with cleaved EYFP-PTHR1 was harvested and were pulled down with ChromoTek-Trap Magnetic Agarose beads (ChromoTek) according to the manufacturer's instruction. Loaded ChromoTek-Trap Magnetic Agarose beads were resuspended in 400 $\mu$l 50 mM ABC Buffer (ammonium bicarbonate). A final concentration of 10 mM DTT was added to each sample for 0.5 h at 37C to perform the reduction step. Addition of a 50 mM iodoacetamide for 0.5 h in the dark at RT followed. Each pull downed and purified transfected cells were digest separately with following proteases: 1.0 $\mu$g Trypsin (#V511C; Promega) overnight at 37°C; 1,5 $\mu$g LysC (#129-02541; Wako Chemicals) overnight at 37°C; 1.5 $\mu$g Chymotrypsin (#V106A; Promega) overnight at 25°C and 1.5 $\mu$g AspN (#11054589001; Roche) overnight at 37°C. The next day the digested peptide mixtures acidified using formic acid (pH < 3) were cleaned and concentrated with StageTips. Peptides were eluted with 50 $\mu$l Acetonitrile (ACN)/Formic acid (FA) (50%/0.1%). The samples were dried in speedvac to remove all the solvents. The samples were resuspended in buffer A (0.1% formic acid and 5% acetonitrile).

For mass spectrometric measurement tryptic samples were online separated and analyzed by reverse phase chromatography on a Vanquish Neo UHPLC system (Thermo Fisher Scientific) in a 44 min non-linear gradient from 2% to 60% Buffer B (80% acetonitrile, 3% formic acid) on a 20 cm self-packed (ID 75 $\mu$m, Dr. Maisch 1.9 $\mu$m AquaBeats) column coupled to an Exploris 480 Orbitrap System (Thermo Fisher Scientific) via electrospray ionization (ESI) and a Top20 DDA acquisition scheme in positive mode. Standard mass spec

setting have been used, which were briefly for MS1: 60 k Resolution, IT 10 ms, AGC Target 300%, 350–1,600 m/z Scan Range, RF-Lens 55%; for MS2 Settings: 15 k Resolution, 1.3 m/z Isolation Window, IT 10 ms, AGC 100%, 28% HCD CE, RF-Lens 50%, 2–6 ChargeState Filter, 20 s dynamic exclusion, Intensity Threshold Filter of 50 k.

For spectrum comparison of PTHR1 specific peptides, raw-files were analyzed using MaxQuant v1.6.7.0 (Mann et al, 2008) for each protease digest separately, setting the corresponding protease as a peptide filter for a semi-specific search (AspN, Trypsin/P, Chemotrypsin and LysC). In this way, non-canonical n-terminal or c-terminal peptides can be identified, as are expected to be generated by ADAM specific cleavage of PTHR-1. Otherwise standard settings were applied: Uniprot Human Database (v2018) as a protein sequence database; Cystein carbamidomethylation as fixed modification and methionine oxidation and n-terminal acetylation as variable modifications. Further data analysis was done in R using the ggplot2, data.table, pracma, cowplot and ggrepel packages. PTHR1-specific peptides were filtered for new n-terminal or c-terminal fragments. Only at position 64/65 both n- and c-terminal specific peptides (and "EVLQRPASIME" and "SDKGWTSASTSGKPRK") could be identified in the cell line with ADAM and PTHR-1 over-expression. For further validation of these peptide identification events, the processed fragment spectra were extracted from andromeda peak list files and compared to predicted peptide spectra generated by the MS2PIP online tool using the default HCD model (Declercq et al, 2023). Predicted Spectra were charge deconvoluted by summing up the intensities of the same fragments but with different charge. Similarity of acquired and predicted spectra was confirmed by calculating the spectrum contrast angle (SCA) (Toprak et al, 2014) from the eight most intense matching fragment peaks (predicted spectrum). For instances an SCA value of 1 would indicate identical spectra, while 0 would indicate absolute disagreement. For each sequence the stability of the SCA was evaluated using a bootstrap approach (n = 20). The corresponding SCA distribution was plotted as a boxplot.

### Global proteome comparison

U2OS cells were seeded at a density of $3 \times 10^6$ cells per 6 cm dish 24 h prior to transfection. Next day, cells in each well were transfected with 0.7 µg PTHR1 and 1.4 µg ADAM19 or empty pcDNA3 vector plasmids. Combinations were transfected with Effectene (QIAGEN) according to the manufacturer's protocol with a total of 2.8 µg of cDNA and cultured for another 48 h. Cells were scraped in RIPA buffer (150 mM NaCl, 50 ml Tris–HCL pH 7.8 in PBS, 0.5% sodium deoxy 0.1% SDS) supplemented with protease and phosphatase inhibitors (Complete and PhosSTOP, Roche Diagnostics) and lysed by use of 26G needles (Sterican; Braun). Lysates were cleared by centrifugation 21,250g, 15 min, 4°C and protein concentration was determined with BCA protocol (Thermo Fisher Scientific). 20 µg of protein was then taken from each sample to perform the SP3 protocol. A final concentration of 10 mM DTT was added to each sample for 1 h at 37°C to perform the reduction step. Addition of a 20 mM iodoacetamide for 45 min in the dark at RT followed. Beads were pipetted into each sample at a concentration of 20 µg/µl from a one to one mix of Sera-Mag SpeedBeads A (45152105050250) and Sera-Mag SpeedBeads B (65152105050250). Then 100% ACN was added to a final concentration of >70% ACN and incubated for 20 min at RT and 1,000 rpm with shaking (Eppendorf Centrifuge 5425/5425 R). Beads with proteins were washed three times with ethanol (80%) and then digested in 100 µl 50 mM ABC buffer (ammonium bicarbonate) with 1.0 µg trypsin (#V511C; Promega) and 1.0 µg LysC overnight at 37°C in the Thermomix at 1,200 rpm (Eppendorf Centrifuge 5425/5425 R). The next day the digested proteins in the supernatant were acidified using formic acid (FA). The samples were dried in speedvac to remove all the solvents. The samples were resuspended in buffer A (0.1% formic acid and 5% acetonitrile). Peptides were separated using reversed-phase liquid chromatography (EASY nLC II 1200; Thermo Fisher Scientific) with self-made C18 microcolumns (20 cm long) packed with ReproSil-Pur C18-AQ 1.9 µm resin (cat# r119.aq.0001; Dr. Maisch). The chromatography system was coupled online to the electrospray ion source (Proxeon) of an Orbitrap Exploris 480 mass spectrometer (Thermo Fisher Scientific). Buffer A (0.1% formic acid, 5% acetonitrile) and buffer B (0.1% formic acid, 80% acetonitrile) made the mobile phase of the chromatography. By increasing the percentage of buffer B, with a flowrate of 250 nl/min, over the course of 110 min, the peptides were eluted from the column and loaded into the mass spectrometer. Mass spectrometry data was acquired in data-independent mode with settings for one full scan (resolution: 120,000; m/z range: 350–1,650; normalized AGC target: 300%; maximum injection time: 20 ms), followed by MS/MS scans (resolution: 30,000; isolation window [m/z]:2, normalized AGC target [%]: 3,000; first mass: 200; HCD collision energies [%]: 26, 29, 32).

Raw files were analyzed with DIANN 1.8.1beta16 (Demichev et al, 2020) in library free method. The REPORT data table was further processed and analyzed using R. The table was filtered for those proteins with Q-value ≤ 0.01, Protein. Q-value ≤ 0.01, mass evidence > 0.5 and precursor charge > 1. The data were then $\log_2$ transformed and the missing values were imputed by withdrawing random values from a generated normal distribution calculated as 0.25 times the SD of the measured log-transformed values, down-shifted by 1.8 SD. A t test, with Benjamin Hochberg multiple comparison testing FDR correction, was employed to compare PTHR1 and PTHR1 + ADAM19 cells lines. As significance cut off levels, we decided on a $\log_2$ fold change > 2 and an adjusted P-value < 0.5. To compare the intensity profiles of the determined significant proteins we took the median intensities for each proteins in each experiment (cell line) and z scored the intensity. The z-scored intensities were plotted in a heatmap (Perseus v1.6.7.0) to compare the profiles.

### Statistics

The experiments were reproduced as indicated in the figure legends. Statistically significant differences were determined by one-way and two-way ANOVA and Bonferroni multi-comparison, t test or log-rank (Mantel-Cox) test. Fiducial limits are given as mean ± SEM. P < 0.05 was accepted as significant.

# Supplementary Information

# Acknowledgements

We thank the members of the family that enabled this study. We fondly remember Prof. Jennifer D Cartwright, consummate physician and Professor of Pediatrics, University of the Witwatersrand, Johannesburg, South Africa, who introduced us to this family. Bärbel Pohl and Astrid Mühl provided excellent technical assistance. Franz Rüschendorf supported the linkage study.

## Author Contributions

A Aydin: data curation, investigation, methodology, and writing—original draft.

C Klenk: formal analysis, investigation, methodology, and writing—review and editing.

K Nemec: investigation, methodology, and writing—review and editing.

A Işbilir: data curation and investigation.

LM Martin: data curation, investigation, and methodology.

H Zauber: investigation and methodology.

T Rrustemi: investigation and methodology.

HR Toka: investigation.

H Schuster: investigation.

M Gong: investigation.

S Stricker: investigation.

A Bock: supervision, investigation, and methodology.

S Bähring: investigation.

M Selbach: supervision and methodology.

MJ Lohse: supervision and investigation.

FC Luft: conceptualization, data curation, formal analysis, supervision, funding acquisition, investigation, and writing—original draft, review, and editing.

## Conflict of Interest Statement

The authors declare that they have no conflict of interest.

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
