## [Reviewer comments · Life Science Alliance]

Life Science Alliance

ADAM19 cleaves the PTH receptor and associates with brachydactyly type E

Atakan Aydin, Christoph Klenk, Katarina Nemeč, Ali İşbilir, Lisa Martin, Henrik Zauber, Trendelina Rrustemi, Hakan Toka, Herbert Schuster, Maolian Gong, Sigmar Stricker, Andreas Bock, Sylvia Bähring, Matthias Selbach, Martin Lohse, and Friedrich Luft

DOI: <https://doi.org/10.26508/lsa.202302400>

Corresponding author(s): Friedrich Luft, Charité - Universitätsmedizin Berlin

Review Timeline:

Submission Date:	2023-09-27
Editorial Decision:	2023-11-13
Revision Received:	2023-12-23
Editorial Decision:	2024-01-19
Revision Received:	2024-01-25
Accepted:	2024-01-25

Transaction Report:

November 13, 2023

Re: Life Science Alliance manuscript #LSA-2023-02400-T

Friedrich C Luft

Experimental and Clinical Research Center (ECRC), a joint cooperation between the Charité Medical Faculty and the Max Delbrück Center for Molecular Medicine (MDC)

Genetics of Disease

Medical Faculty of the Charite Humboldt University of Berlin Experimental and Clinical Research Center

Franz-Volhard Clinic

Berlin 13125

Germany

Dear Dr. Luft,

Thank you for submitting your manuscript entitled "ADAM19 cleaves PTHR1 and associates with brachydactyly type E" to Life Science Alliance. The manuscript was assessed by expert reviewers, whose comments are appended to this letter. We invite you to submit a revised manuscript addressing the Reviewer comments.

Thank you for this interesting contribution to Life Science Alliance. We are looking forward to receiving your revised manuscript.

Sincerely,

Eric Sawey, PhD

Executive Editor

Life Science Alliance

<http://www.lsjournal.org>

B. MANUSCRIPT ORGANIZATION AND FORMATTING:

Reviewer #1 (Comments to the Authors (Required)):

This manuscript describes a family with brachydactyly type E (BDE) in whom the authors identified a complex mutation in the ADAM19 gene segregating with the affected family members. They further characterized the function of ADAM19 regarding the PTH receptor, which is critical in growth plate chondrocyte differentiation and is known to be involved in the pathogenesis of BDE.

This is a well-written manuscript, and the study is well-designed. Overall, the genetic findings are robust. The functional data is also valuable, but the role of ADAM19 regarding the action of PTHrP in the growth plate and how the heterozygous loss of ADAM19 protein causes disease remains unclear. Some results require careful discussion of how they should be interpreted.

I have the following points:

- 1) Data strongly support that the mutation affects protein formation and function. However, in the first paragraph of page 6, the authors wrote: "... and verified premature termination of mutated ADAM19." This statement is too strong for the presented findings. The Western blot data indicates no evidence of the mutant protein, even when using the Y 50-122 antibody. Did the authors run SDS-PAGE with a higher polyacrylamide concentration to see if they can detect a mutant protein? Or does the antibody not recognize the mutant protein due to the changes in the amino acid sequence located within the antigenic epitope? Alternatively, it is probable that the mutation results in nonsense-mediated RNA decay and, therefore, a loss of protein. The statement that the data verified premature termination should be revised to include these caveats.
- 2) Data in Figures 6D and 6E: The data is strong in terms of showing that the (56-63)A PTH1R mutant acts in a different way in terms of Gs and Gq signaling. However, there is a difference between the effects of PTH and PTHrP in terms of Gs activation, which is observed with both the wild-type receptor and the (56-63)A mutant. The authors should discuss why these two ligands for the same receptor are affected in a distinct manner in those experiments and whether this difference may have any implication for BDE pathogenesis. Also, such a difference does not seem to exist in the FRET-based cAMP assays presented in Figure 7. The authors should highlight the difference between BRET vs. FRET-based findings concerning the effects of PTH vs PTHrP and revise the Discussion section to explain these discrepancies.
- 3) The mass-spectrometry proteomic findings obtained from U2OS cells are highly valuable, hinting at the physiologic and disease-related mechanisms governing ADAM19's role in PTH1R signaling. Despite their potential importance, the authors did not confirm these proteomic screen-based results independently, at least for a set of potentially relevant proteins. That way, the reproducibility of these alterations would be shown, providing a strong foundation for subsequent studies aiming to interrogate their role in the action of ADAM19 and BDE pathogenesis.
- 4) Furthermore, do the authors attribute the differential protein expression between PTH1R overexpressing U2OS cells and those co-expressing PTH1R and ADAM18 to altered signaling at baseline? The authors do not seem to have treated the cells with PTH or PTHrP before harvesting cells for the proteomic analysis. That would be more relevant as the functional experiments with Gs, Gq, and arrestin recruitment do not suggest a significant difference at basal levels. In addition, U2OS cells are derived from osteosarcoma and, therefore, may not reflect the chondrocytes where the mechanism involving the interaction between ADAM19 and PTH1R is predicted to occur. These caveats should be discussed.

Additional comments:

- 1) Data in Figure 5B: The mutants significantly blunted the N-terminal shedding by wild-type ADAM19 but did not entirely abolish it. How do the authors explain this finding?
- 2) Page 8, first line: The authors should explain why they chose CD4 as a control. As controls, it would also have been valuable to use a different protease with PTH1R, hoping to have a specific effect of ADAM19. In addition, the authors could consider expressing wild-type ADAM19 with another GPCR to determine whether ADAM19's effect is specific to PTH1R.

- 3) Supplementary Fig 1 and its legend are unclear: The text in the first paragraph of Results says " ... and detected six genomic regions with LOD score above 1.5." However, Panel A in this figure shows more than 6 peaks with LOD scores higher than 1.5. Could the authors explain which 6 were considered here and why the remaining ones were not? The authors should revise the text to reflect the data in the figure correctly or label the figure to explain the details. Also, it is unclear whether this panel refers to the SNP-based assay's results or the microsatellite marker-based fine-mapping. This should be clarified.
- 4) Data in Supplemental Figure 4: The cell morphology seems rounder upon the expression of mutant ADAM19 with PTHR1 than PTHR1 alone, as suggested by PTHR1 immunofluorescence. Is this consistently observed? Since PTHR1 immunofluorescence outlines the cell morphology, one wonders whether the distribution of PTHR1 expression differs according to the presence of functional ADAM19. Did the authors use a plasma membrane marker separately to distinguish this possibility?
- 5) Last line on page 10: When the authors say, "cells expressing PTHR1", I suppose the authors mean over-expressing, considering that U2OS cells have endogenous PTH1R. Is that correct?

Reviewer #2 (Comments to the Authors (Required)):

This interesting report identifies a new mutation in ADAM19 as a new likely cause of brachydactyly type E (BDE) in a family in which the phenotype is transmitted as an autosomal dominant condition. The authors perform genetic analyses, which establish the mutation to be a disruption in the ADAM19 gene predicted to result in a truncated and inactive protein. Based on prior published work from this group showing that the PTH1 receptor can be cleaved by matrix metalloproteinases (MMPs), of which ADAM19 is one, and their other reports linking BDE to defects in the PTHrP/PTH1R signaling system, the authors hypothesize that ADAM19 cleaves the PTH1R and hence that the patient BDE arises from impaired PTH1R cleavage during bone development. They test the hypothesis by performing an extensive series of cell-based experiments that includes mass-spec analysis of cleaved receptor fragments, and different functional assays of effector and G protein coupling. The data provide evidence that ADAM19, but not an inactivate mutant, can cleave the PTH1R, specifically at E64/S65 in the first extracellular loop, and that cleavage results in altered receptor signaling properties in response to PTH or PTHrP ligands, as it appears to enhance Gs activation and barrestin recruitment in response to PTH, but not PTHrP. They suggest that the patient phenotype involves such alterations in PTH1R cleavage and signaling responses during bone development.

The study is overall, well presented and poses some new and potentially important biology for bone development and the PTH1R. The genetic analyses seem robust and convincing that the identified ADAM19 mutation causes the BDE phenotype. The in vitro and biochemical data on PTH1R cleavage and functional effects are also generally convincing, although some effects seem small and not supported by statistic. The model that the ADAM19 mutation results in a change in PTH1R cleavage and signaling in vivo to contribute to the patient phenotype also seems plausible, given the new data, and raises questions that can be pursued in future research.

Specific comments.

- 1) The authors show that PTH1R cleavage by ADAM19 results in an enhanced Gs-activation response to PTH, which agrees with the enhanced PTH-induced cAMP response data they reported previously (Klenk et al. 2022). That the patient mutation would result in a blunting of the PTH1R cAMP response, albeit to PTHrP, via reduced receptor cleavage, also seems consistent with BDE being associated in other cases with reduced PTHrP/PTH1R mediated cAMP signaling. The more recent study by Chu et al, however, (eLife, 2023; <https://doi.org/10.7554/eLife.82142>) provides data showing that cleavage of the PTH1R in cells by MMP14 blunts the cAMP response to PTH, and that genetic ablation of MMP14 in bone cells in mice enhances the proliferative and anabolic responses of the cells to injected PTH ligands, which suggests that PTH1R cleavage down-regulates PTH1R signaling, at least in this bone system with PTH ligand. It would be useful if the authors could comment on how their findings relate to those in the Chu et al study, as the results seem somewhat contradictory.
- 2) Figure 6 and 8. An indication of significance in the differences between with vs. without ADAM19 cleavage for the Gs, GQ (Fig 6) and barrestin (Fig 8) responses is needed?
- 3) It would be interesting to know more about a potential mechanism for the differential effects of cleavage on PTH vs PTHrP interaction. For example, are there amino acid sequence differences in the relevant regions of PTH and PTHrP that could result in altered interaction with the tethered helix1 of the cleaved receptor (as discussed in Klenk et al 2022)? A comment here could help in interpreting the data and understanding mechanism.
- 4) Figure 7. How can the apparent discrepancy between lack of effect of ADAM19 cleavage on the cAMP response to PTH in U2OS cells (Fig 7) and the enhancement in the GS-activation response to PTH (Fig 6 B) be explained? Could this involve the apparent enhanced barrestin coupling with PTH on the cleaved receptor (Fig 8A and Supplemental Fig 10B)? Could the differences in the cell types and/or transfection efficiencies used for the two assays be a factor? Were cAMP responses assessed in HEK293 or SaOS2 cells?
- 5) In Figure 6, the control for cleavage is the use of a non-cleavable PTH1R-Ala mutant (vs PTHR WT), whereas in Fig 7 the control is co-transfecting PTH1R without ADAM19 (vs with). Is there a reason why the same controls could not be used for each of the assays? A further control would be to assess the inhibitors, batimastat or Ilomastat, for effect in these signaling

responses--has this been done?

6) Based on the mass spec data of Fig 5C, the cleavage site is pinpointed to Glu64, while in Klenk et al 2022, Ser61 is identified as the primary MMP cleavage site, with secondary exopeptidase sites at Ser65, Ser73 and Lys80. Could Ser61 also be the primary site for ADAM19 cleavage and E64 be a secondary site? Can protein prediction analysis tools help confirm the ADAM19 cut site for PTH1R?

7) Supplemental Figure 10B barrestin recruitment in SaOS2 cells. The legend needs to identify what the green and black traces are in terms of how the cells were transfected. As these cells are well known to express endogenous PTH receptors and produce robust cAMP responses to PTH ligands, it would be interesting to know whether ADAM19 alters responses via the endogenous receptor in these cells--has this been evaluated?

8) Supplemental Fig 8. Check that data are from Figure 8 and not from Figure 7 as indicated.

9) Supplemental Fig 12. For panel A legend--the y axis needs to be defined. Please check that the X-axis label correct--i.e. that the ratio is not inverted. For Panel B, the four cell types need to be explained in the legend.

10) it would be worth noting whether any dental phenotype is observed in the affected family since many heterozygous L-O-F PTH1R mutations have been reported for patients with tooth eruption defects.

11) Page 11 last sentence and continued on page 12. Check that the "The cleaved receptor exhibited reduced Gs activation..." is correct; the data seem to show enhanced Gs activation with cleavage.

12) The terms "ADAM19-EA" in Fig. 3 legend and "ADAM-MT" in Results, page 7 need to be defined.

Discussion: page 15, last sentence of paragraph 2 "Independent of our findings, ADAM19..." is ambiguous and needs clarification. Page 16 first paragraph last sentence, the term should be "...receptor activity modifying protein 2..."; page 17, the relevance of the sentence regarding LGR6 is unclear. Page 17, last sentence: "...not necessarily circumstantial..." is unclear; "...is circumstantial..." seems simpler, and more precise.

Reviewer #1 (Comments to the Authors (Required)):

This manuscript describes a family with brachydactyly type E (BDE) in whom the authors identified a complex mutation in the ADAM19 gene segregating with the affected family members. They further characterized the function of ADAM19 regarding the PTH receptor, which is critical in growth plate chondrocyte differentiation and is known to be involved in the pathogenesis of BDE.

This is a well-written manuscript, and the study is well-designed. Overall, the genetic findings are robust. The functional data is also valuable, but the role of ADAM19 regarding the action of PTHrP in the growth plate and how the heterozygous loss of ADAM19 protein causes disease remains unclear. Some results require careful discussion of how they should be interpreted.

We appreciate the reviewer's positive comments and are pleased that our genetic arguments have passed muster.

I have the following points:

1) Data strongly support that the mutation affects protein formation and function. However, in the first paragraph of page 6, the authors wrote: "... and verified premature termination of mutated ADAM19." This statement is too strong for the presented findings. The Western blot data indicates no evidence of the mutant protein, even when using the Y 50-122 antibody. Did the authors run SDS-PAGE with a higher polyacrylamide concentration to see if they can detect a mutant protein? Or does the antibody not recognize the mutant protein due to the changes in the amino acid sequence located within the antigenic epitope? Alternatively, it is probable that the mutation results in nonsense-mediated RNA decay and, therefore, a loss of protein. The statement that the data verified premature termination should be revised to include these caveats.

The claim has been modified. We include (now) on page 7, the explanation outlined below: Mutation of ADAM19 leads to a frame shift resulting in a scrambled sequence after His116 and a new stop codon after amino acid 135. The western blot experiments with antibodies against residues 50-122 of ADAM19 or with antibodies against the Flag epitope, which had been fused to the native C-terminus of ADAM19, suggest that neither the N-terminal fragment nor full-length ADAM19 (e.g. as a result from a read-through event of the new stop codon) have formed and thus defective ADAM19 possibly has been degraded in the cell. However, we cannot fully exclude that the absence of ADAM19 at protein level may also be due to a pre-translational event. (as suggested by the reviewer) Therefore, we have modified our statement.

2) Data in Figures 6D and 6E: The data are strong in terms of showing that the (56-63)A PTH1R mutant acts in a different way in terms of Gs and Gq signaling. However, there is a difference between the effects of PTH and PTHrP in terms of Gs activation,

which is observed with both the wild-type receptor and the (56-63)A mutant. The authors should discuss why these two ligands for the same receptor are affected in a distinct manner in those experiments and whether this difference may have any implication for BDE pathogenesis. Also, such a difference does not seem to exist in the FRET-based cAMP assays presented in Figure 7. The authors should highlight the difference between BRET vs. FRET-based findings concerning the effects of PTH vs PTHrP and revise the Discussion section to explain these discrepancies.

We have added caveats (now) on page 18 in the discussion (as summarized below): The observed differences in PTH and PTHrP effects on PTH1R-related Gs activation could potentially stem from variations in the execution time points of the assays (different days and different passage of the cells). Our observations suggest that the dynamic range of the Gs BRET assay exhibits greater variability compared to the Gq BRET assay, that was more stable and had lower day-to-day variability. Alternatively, the observed differences could arise from the specific timepoint chosen for generating the concentration-response curve, as PTHrP-mediated Gs activation reaches saturation at a distinct point compared to PTH-mediated activation. FRET experiments, which measure cAMP accumulation, inherently evaluate processes further downstream in the signaling cascade. This state-of-affairs could introduce the possibility of other factors masking the observed effects closer to the receptor, such as enhanced beta-arrestin interactions (as seen in Figure 8). Additionally, the transient transfection of ADAM19 with incomplete cleavage (as seen in Figure 5B), different cell lines used (HEK293 vs U2OS), and different heterologous expression protocol (transfection vs transduction) all could potentially obscure downstream effects of enhanced Gs signaling.

The reason for the different Gs/Gq-activation profiles stimulated by either PTH or PTHrP likely stems from different active receptor conformations stabilized by the two ligands. Also, the maximal BRET change in G proteins is smaller with PTHrP, making ligand differences more difficult to interpret. The apparent difference the Reviewer mentions could stem from the small experimental window in PTHrP measurements. Given the little difference between the two agonists, it is not surprising that there is no difference in cellular cAMP production stimulated by either PTH or PTHrP.

3) The mass-spectrometry proteomic findings obtained from U2OS cells are highly valuable, hinting at the physiologic and disease-related mechanisms governing ADAM19's role in PTH1R signaling. Despite their potential importance, the authors did not confirm these proteomic screen-based results independently, at least for a set of potentially relevant proteins. That way, the reproducibility of these alterations would be shown, providing a strong foundation for subsequent studies aiming to interrogate their role in the action of ADAM19 and BDE pathogenesis.

Admittedly, the proteomics data are preliminary. We agree with the reviewer that it would be interesting and insightful to validate some of the proteins identified in the global proteome analysis. We present these data as preliminary. A more extensive validation would necessitate extensive

experimental work, which is beyond the scope of our current study. Moreover, it's crucial to emphasize that the primary objective of our global proteome analysis was to identify any alterations in protein abundance across different cell lines and to ascertain the impact of mutations on various biological processes. While we acknowledge that we did not validate the individual proteins, we firmly believe that our data holds significant value for the scientific community and will be a useful resource for future research in this area. We have made some adjustments in the legend to the proteomic data (now Supplement Figure 13).

4) Furthermore, do the authors attribute the differential protein expression between PTH1R overexpressing U2OS cells and those co-expressing PTH1R and ADAM18 to altered signaling at baseline? The authors do not seem to have treated the cells with PTH or PTHrP before harvesting cells for the proteomic analysis. That would be more relevant as the functional experiments with Gs, Gq, and arrestin recruitment do not suggest a significant difference at basal levels. In addition, U2OS cells are derived from osteosarcoma and, therefore, may not reflect the chondrocytes where the mechanism involving the interaction between ADAM19 and PTH1R is predicted to occur. These caveats should be discussed.

We concur that the use of U2OS cells is perhaps artificial (now acknowledged on page 11). Given that both cell types originate from osteoprogenitor cells, we hypothesized that the U2OS cell line would exhibit a molecular background of relevant interacting proteins more closely resembling those of chondrocytes, making it the most suitable model cell system available for our investigation.

Additional comments:

1) Data in Figure 5B: The mutants significantly blunted the N-terminal shedding by wild-type ADAM19 but did not entirely abolish it. How do the authors explain this finding?

Our extensive response is outlined on page 20. The transfection of ADAM19 was transient. As we and others have shown earlier (c.f. Klenk et al., 2010; Chu et al., 2023), PTH1R-ECD can also be subject to cleavage by proteases other than ADAM19. Indeed, we observe a minor fraction of processed PTH1R that may be cleaved by other metalloproteases endogenously expressed in HEK293 cells - the cell line that had been employed for the experiments (see also our response to comment 1 of reviewer 2).

2) Page 8, first line: The authors should explain why they chose CD4 as a control. As controls, it would also have been valuable to use a different protease with PTH1R, hoping to have a specific effect of ADAM19. In addition, the authors could consider

expressing wild-type ADAM19 with another GPCR to determine whether ADAM19's effect is specific to PTH1R.

We selected CD4 as a control in our assay due to its characteristics as a single transmembrane protein, akin to our experimental protein ADAM19. Yet, CD4 lacks the shedding activity. This choice was strategic to serve as a negative control in our shedding assay. By using a control that inherently does not exhibit shedding activity, we aimed to delineate the absence of shedding, thereby validating the specificity of our assay for detecting protease-mediated shedding of PTH1R, with particular focus on ADAM19 (insert added on page 9).

Regarding the suggestion to use a different protease with PTH1R, we acknowledge the merit of the reviewer's suggestion. It indeed could have provided additional insights into the specific effect of ADAM19. However, due to resource limitations, we were unable to include a broader range of proteases in this study. Moreover, our aim was not to characterize a panel of proteases concerning PTH1R shedding, but rather specific to addressing the functional interaction between ADAM19 and PTH1R. Nevertheless, this is a valuable direction for future research as it could provide valuable insights in PTH1R modulation via different proteases.

As for expressing wild-type ADAM19 with another GPCR to determine its effect specificity to PTH1R, we understand the potential implications of such an experiment. However, our study was primarily focused on the interaction between ADAM19 and PTH1R. Considering the limited understanding of which GPCRs interact with proteases like ADAM19, introducing another GPCR could introduce significant variables and risk concealing the specific objectives of our study. Nevertheless, we acknowledge the scientific value in investigating the interaction of ADAM19 with other GPCRs and this aim could be considered for future research.

3) Supplementary Fig 1 and its legend are unclear: The text in the first paragraph of Results says " ... and detected six genomic regions with LOD score above 1.5." However, Panel A in this figure shows more than 6 peaks with LOD scores higher than 1.5. Could the authors explain which 6 were considered here and why the remaining ones were not? The authors should revise the text to reflect the data in the figure correctly or label the figure to explain the details. Also, it is unclear whether this panel refers to the SNP-based assay's results or the microsatellite marker-based fine-mapping. This should be clarified.

The reviewer is correct. Corrections have been incorporated on page 6 and elsewhere as appropriate. Our text here was not clear (ie. erroneous). When we first examined this family, in the mid 1990's, the genome had not been defined. When SNP genotyping became available, we (largely) generated the data shown in Supplemental figure 1. We then worked up additional microsatellite markers to improve the results and ended up with Supplemental figure 1. We then (when technology became available) relied on total-genome sequencing in affected and non-affected persons. The sole abnormality that we found with appropriate inheritance features was the ADAM19 mutation on chromosome 5.

4) Data in Supplemental Figure 4: The cell morphology seems rounder upon the expression of mutant ADAM19 with PTHR1 than PTHR1 alone, as suggested by PTHR1 immunofluorescence. Is this consistently observed? Since PTHR1 immunofluorescence outlines the cell morphology, one wonders whether the distribution of PTHR1 expression differs according to the presence of functional ADAM19. Did the authors use a plasma membrane marker separately to distinguish this possibility?

These experiments were performed on separate days and perhaps the confluence was not precisely the same in all the experiments. This state-of-affairs could perhaps explain why the morphology of the cells appears differently in the data presented.

5) Last line on page 10: When the authors say, "cells expressing PTHR1", I suppose the authors mean over-expressing, considering that U2OS cells have endogenous PTH1R. Is that correct?

Indeed, we meant overexpressing (now changed). Over has been added now page 13.

Reviewer #2 (Comments to the Authors (Required)):

This interesting report identifies a new mutation in ADAM19 as a new likely cause of brachydactyly type E (BDE) in a family in which the phenotype is transmitted as an autosomal dominant condition. The authors perform genetic analyses, which establish the mutation to be a disruption in the ADAM19 gene predicted to result in a truncated and inactive protein. Based on prior published work from this group showing that the PTH1 receptor can be cleaved by matrix metalloproteinases (MMPs), of which ADAM19 is one, and their other reports linking BDE to defects in the PTHrP/PTH1R signaling system, the authors hypothesize that ADAM19 cleaves the PTH1R and hence that the patient BDE arises from impaired PTH1R cleavage during bone development. They test the hypothesis by performing an extensive series of cell-based experiments that includes mass-spec analysis of cleaved receptor fragments, and different functional assays of effector and G protein coupling. The data provide evidence that ADAM19, but not an inactivate mutant, can cleave the PTH1R, specifically at E64/S65 in the first extracellular loop, and that cleavage results in altered receptor signaling properties in response to PTH or PTHrP ligands, as it appears to enhance Gs activation and beta-arrestin recruitment in response to PTH, but not PTHrP. They suggest that the patient phenotype involves such alterations in PTH1R cleavage and signaling responses during bone development.

The study is overall, well presented and poses some new and potentially important biology for bone development and the PTH1R. The genetic analyses seem robust and convincing that the identified ADAM19 mutation causes the BDE phenotype. The in

vitro and biochemical data on PTH1R cleavage and functional effects are also generally convincing, although some effects seem small and not supported by statistic. The model that the ADAM19 mutation results in a change in PTH1R cleavage and signaling in vivo to contribute to the patient phenotype also seems plausible, given the new data, and raises questions that can be pursued in future research.

We thank the reviewer for the encouraging assessment and will do our best to make the presentation more robust.

Specific comments.

1) The authors show that PTH1R cleavage by ADAM19 results in an enhanced Gs-activation response to PTH, which agrees with the enhanced PTH-induced cAMP response data they reported previously (Klenk et al. 2022). That the patient mutation would result in a blunting of the PTH1R cAMP response, albeit to PTHrP,, via reduced receptor cleavage, also seems consistent with BDE being associated in other cases with reduced PTHrP/PTH1R mediated cAMP signaling. The more recent study by Chu et al, however, (eLife, 2023; <https://doi.org/10.7554/eLife.82142>) provides data showing that cleavage of the PTH1R in cells by MMP14 blunts the cAMP response to PTH, and that genetic ablation of MMP14 in bone cells in mice enhances the proliferative and anabolic responses of the cells to injected PTH ligands, which suggests that PTH1R cleavage down-regulates PTH1R signaling, at least in this bone system with PTH ligand. It would be useful if the authors could comment on how their findings relate to those in the Chu et al study, as the results seem somewhat contradictory.

We have incorporated these important findings into our discussion (manuscript page 21). The work by Chu et al. has been duly cited. Thank you for this sage advice.

2) Figure 6 and 8. An indication of significance in the differences between with vs. without ADAM19 cleavage for the Gs, GQ (Fig 6) and b-arrestin (Fig 8) responses is needed?

Significance findings have been added.

3) It would be interesting to know more about a potential mechanism for the differential effects of cleavage on PTH vs PTHrP interaction. For example, are there amino acid sequence differences in the relevant regions of PTH and PTHrP that could result in altered interaction with the tethered helix1 of the cleaved receptor (as discussed in Klenk et al 2022)? A comment here could help in interpreting the data and understanding mechanism.

This is an interesting question, and we have now added a new paragraph to the discussion (page 15-16) commenting on the structural aspects of our findings along with a new Supplemental Figure 12. The purpose of this figure is to demonstrate interaction of PTH and PTHrP with the extracellular domain of PTH1R.

4) Figure 7. How can the apparent discrepancy between lack of effect of ADAM19 cleavage on the cAMP response to PTH in U2OS cells (Fig 7) and the enhancement in the GS-activation response to PTH (Fig 6 B) be explained? Could this involve the apparent enhanced barrestin coupling with PTH on the cleaved receptor (Fig 8A and Supplemental Fig 10B)? Could the differences in the cell types and/or transfection efficiencies used for the two assays be a factor? Were cAMP responses assessed in HEK293 or SaOS2 cells?

We did our best to deal with this point on page 20-21 in the discussion. The points raised are valid – increased b-arrestin coupling could counteract the observed rise in Gs activation and mask increased cAMP accumulation. Additionally, the choice of cell type (HEK293 vs U2OS) and protocols for heterologous expression could impact the outcome, as the stoichiometry and expression levels of interacting proteins can vary. We believe that all issues raised by the reviewer could be operative. ADAM19 cleavage could be incomplete. Cell-type models are confounding variables.

5) In Figure 6, the control for cleavage is the use of a non-cleavable PTH1R-Ala mutant (vs PTHR WT), whereas in Fig 7 the control is co-transfecting PTH1R without ADAM19 (vs with). Is there a reason why the same controls could not be used for each of the assays? A further control would be to assess the inhibitors, batimastat or I lomastat, for effect in these signaling responses--has this been done?

We concede that a more consistent approach would have been preferable. These differences are largely logistical phenomena (different laboratories, different time points, different resources).

6) Based on the mass spec data of Fig 5C, the cleavage site is pinpointed to Glu64, while in Klenk et al 2022, Ser61 is identified as the primary MMP cleavage site, with secondary exopeptidase sites at Ser65, Ser73 and Lys80. Could Ser61 also be the primary site for ADAM19 cleavage and E64 be a secondary site? Can protein prediction analysis tools help confirm the ADAM19 cut site for PTH1R?

Computational analysis of the PTH1R-ECD for protease cleavage using position weight matrices for 11 MMPs has not revealed evidence for cleavage at Glu64 (c.f. Klenk et al., 2022). However, this situation does not necessarily argue against our current experimental evidence for ADAM19 cleavage at Glu64. Metalloprotease cleavage sites are poorly conserved and hard to predict, and to our knowledge

currently no computational tools exist to predict ADAM protease sites. Perhaps access to services could be helpful in the future.

https://wiki.bits.vib.be/index.php/Prediction_of_protease_cleavage_sites.

7) Supplemental Figure 10B b-arrestin recruitment in SaOS2 cells. The legend needs to identify what the green and black traces are in terms of how the cells were transfected. As these cells are well known to express endogenous PTH receptors and produce robust cAMP responses to PTH ligands, it would be interesting to know whether ADAM19 alters responses via the endogenous receptor in these cells--has this been evaluated?

The legend has been changed accordingly. Unfortunately, the current assay system, which relies on BRET between PTH1R-Nanoluc and B-arrestin2-cpVenus, is incapable of quantifying endogenous receptor-induced B-arrestin2 recruitment. While evaluating this would be highly valuable, it would necessitate the development of a novel assay system for untagged receptors (as used in <https://doi.org/10.1016/j.jbc.2021.100503>).

8) Supplemental Fig 8. Check that data are from Figure 8 and not from Figure 7 as indicated.

Data are from Fig. 8. The legend has been corrected. Please note that the supplemental figure legend has been altered and supplemental figure 8 is now changed to Supplement Figure 11.

9) Supplemental Fig 12. For panel A legend--the y axis needs to be defined. Please check that the X-axis label correct--i.e. that the ratio is not inverted. For Panel B, the four cell types need to be explained in the legend.

The legend and the figure have been corrected. We added the following sentence to the figure legends. "with Log2 fold changes in the x-axis and the Log10 transformed p-values in the y-axis." and "(PTH1R= PTH1R overexpression, ADAM19+PTH1R = ADAM19 and PTH1R overexpression, ADAM19= ADAM19 overexpression and endogenous= no transfection control)". Please note that the supplemental figure legend has been altered and supplemental figure 12 is now changed to Supplement Figure 13.

10) it would be worth noting whether any dental phenotype is observed in the affected family since many heterozygous L-O-F PTH1R mutations have been reported for patients with tooth eruption defects.

Important question. We have observed no specific dental phenotypes in our patients (all BDE3 families. A search of Pub Med also gave a negative result. Pererda et al reported that inactivating PTH/PTHrP signaling disorders (iPPSDs) are associated with BDE3, although they reported no dental abnormalities. We added a sentence regarding phenotypes in the results section.

11) Page 11 last sentence and continued on page 12. Check that the "The cleaved receptor exhibited reduced Gs activation..." is correct; the data seem to show enhanced Gs activation with cleavage.

The Reviewer is correct. Cleaved receptor exhibits enhanced Gs activation. We thank the reviewer for spotting this mistake. We have corrected the text (on page 14) accordingly.

12) The terms "ADAM19-EA" in Fig. 3 legend and "ADAM-MT" in Results, page 7 need to be defined.

Discussion: page 15, last sentence of paragraph 2 "Independent of our findings, ADAM19..." is ambiguous and needs clarification. Page 16 first paragraph last sentence, the term should be "...receptor activity modifying protein 2..."; page 17, the relevance of the sentence regarding LGR6 is unclear. Page 17, last sentence: "...not necessarily circumstantial..." is unclear; "...is circumstantial..." seems simpler, and more precise.

The legend to figure 3 has been altered and enhanced and other modifications made.

January 19, 2024

RE: Life Science Alliance Manuscript #LSA-2023-02400-TR

Prof. Friedrich C Luft
Charité - Universitätsmedizin Berlin
Experimental and Clinical Research Center, a cooperation between the Max Delbrück Center for Molecular Medicine in the
Helmholtz Association and Charité Universitätsmedizin Berl
Lindenberger Weg 80
Berlin, Berlin 13125
Germany

Dear Dr. Luft,

Thank you for submitting your revised manuscript entitled "ADAM19 cleaves PTHR1 and associates with brachydactyly type E". We would be happy to publish your paper in Life Science Alliance pending final revisions necessary to meet our formatting guidelines.

- please address Reviewer 2's remaining comments
- please be sure that the authorship listing and order is correct
- Please upload all figure files as individual ones, including the supplementary figure files; for publication, we require PowerPoint, TIFF, PDF, or EPS files
- please add the Twitter handle of your host institute/organization as well as your own or/and one of the authors in our system
- please note that the titles in the system and on the manuscript file must match
- please remove Graphical Abstract from the manuscript file and upload it separately with the file designation "Graphical Abstract"
- please add your main and supplementary figure legends to the main manuscript text after the references section
- please add callouts for Figures 8A-B; S1C,D; S7A-C; S9A-D; S10A-d; S11A-D; S12A-C; S13A, B to your main manuscript text

A. FINAL FILES:

B. MANUSCRIPT ORGANIZATION AND FORMATTING:

Sincerely,

Reviewer #1 (Comments to the Authors (Required)):

The authors have adequately addressed my criticisms of the original version.

Reviewer #2 (Comments to the Authors (Required)):

The authors have addressed my main concerns and have added on page 15/16 useful new discussion on how cleavage of the PTH1R might impact binding of PTH and PTHrP ligands based on structural models that are now shown in new Supplement Figure 12. I suggest that this section be revised, however, as it indicates the following:

"Proteolytic cleavage of the extracellular domain at position 64 will likely increase the flexibility of the N-terminal α 1 helix, which is located proximal to the cleavage site and curbs on side of the extracellular binding pocket. Given the ligand-specific contacts at the proximal end of α 1 helix, it is well conceivable that the likely increased flexibility of the binding pocket after receptor cleavage may result in the observed ligand-dependent changes in receptor signaling. ",

which seems somewhat incorrect, as the α 1 helix is contained in the segment V31-Q57, as shown by the structural model PDB.6fj3, such that Adam 19 cleavage at Glu64 would result in removal of the helix altogether, rather than an increase in its flexibility. It would therefore seem more correct to indicate that cleavage at Glu64 would result in the loss of the α 1 helix and hence a likely increase in the flexibility of the ECD binding pocking.

Some rewording of this section therefore seems to be needed.

Reviewer #2 (Comments to the Authors (Required)):

The authors have addressed my main concerns and have added on page 15/16 useful new discussion on how cleavage of the PTH1R might impact binding of PTH and PTHrP ligands based on structural models that are now shown in new Supplement Figure 12. I suggest that this section be revised, however, as it indicates the following:

"Proteolytic cleavage of the extracellular domain at position 64 will likely increase the flexibility of the N-terminal α 1 helix, which is located proximal to the cleavage site and curbs on side of the extracellular binding pocket. Given the ligand-specific contacts at the proximal end of α 1 helix, it is well conceivable that the likely increased flexibility of the binding pocket after receptor cleavage may result in the observed ligand-dependent changes in receptor signaling. ",

which seems somewhat incorrect, as the α 1 helix is contained in the segment V31-Q57, as shown by the structural model PDB.6fj3, such that Adam 19 cleavage at Glu64 would result in removal of the helix altogether, rather than an increase in its flexibility. It would therefore seem more correct to indicate that cleavage at Glu64 would result in the loss of the \$\alpha\$ 1 helix and hence a likely increase in the flexibility of the ECD binding pocking.

Some rewording of this section therefore seems to be needed.

Response:

We appreciate the reviewer's concerns and now address the issue. In fact, we found that after proteolytic cleavage the \$\alpha\$ 1 helix remains tethered to the distal ECD through a disulfide bond, which is formed between Cys48 and Cys117. We were able to show the same state-of-affairs in our previous work (c.f. Klenk et al., 2010 and Klenk et al., 2022). Thus, \$\alpha\$ 1 helix is still able to serve ligand binding. We have modified the respective sentence and adjusted Suppl. Fig. 12 for additional clarification.

January 25, 2024

RE: Life Science Alliance Manuscript #LSA-2023-02400-TRR

Prof. Friedrich C Luft
Charité - Universitätsmedizin Berlin
Experimental and Clinical Research Center, a cooperation between the Max Delbrück Center for Molecular Medicine in the
Helmholtz Association and Charité Universitätsmedizin Berl
Lindenberger Weg 80
Berlin, Berlin 13125
Germany

Dear Dr. Luft,

Thank you for submitting your Research Article entitled "ADAM19 cleaves the PTH receptor and associates with brachydactyly type E". It is a pleasure to let you know that your manuscript is now accepted for publication in Life Science Alliance. Congratulations on this interesting work.

DISTRIBUTION OF MATERIALS:

Again, congratulations on a very nice paper. I hope you found the review process to be constructive and are pleased with how the manuscript was handled editorially. We look forward to future exciting submissions from your lab.

Sincerely,
